# CAOS: Conformal Aggregation of One-Shot Predictors

**Maja Waldron** [1]

## Abstract

One-shot prediction enables rapid adaptation of pretrained foundation models to new tasks using only one labeled example, but lacks principled uncertainty quantification. While conformal prediction provides finite-sample coverage guarantees, standard split conformal methods are inefficient in the one-shot setting due to data splitting and reliance on a single predictor. We propose Conformal Aggregation of One-Shot Predictors (CAOS), a conformal framework that adaptively aggregates multiple one-shot predictors and uses a leave-one-out calibration scheme to fully exploit scarce labeled data. Despite violating classical exchangeability assumptions, we prove that CAOS achieves valid marginal coverage using a monotonicity-based argument. Experiments on one-shot facial landmarking and RAFT text classification tasks show that CAOS produces substantially smaller prediction sets than split conformal baselines while maintaining reliable coverage.

## 1. Introduction

Large pretrained foundation models enable rapid adaptation to new tasks using only a handful of labeled examples, without task-specific fine-tuning (Brown et al., 2020; Bommasani, 2021; Radford et al., 2021; Siméoni et al., 2025). In both vision and language, such one-shot and few-shot prediction paradigms are particularly attractive in settings where labeled data are scarce or expensive to obtain.

In these settings, each labeled example can serve as a demonstration that induces a task-specific predictor for new inputs. When multiple labeled examples are available, this naturally gives rise to a *collection* of one-shot predictors rather than a single fixed model. The quality and relevance of these predictors may vary substantially across test inputs, mak-

ing it unclear which prediction to trust and how to quantify uncertainty in a principled way.

A motivating example arises in medical facial analysis, where clinicians annotate a small number of images with task-specific facial landmarks for pre- and post-operative assessment. Modern vision foundation models can transfer landmark information from a single annotated image to a new image via patch similarity (Siméoni et al., 2025), yielding a collection of one-shot predictors whose usefulness varies across patients and poses.

Beyond point predictions, practical deployment in such settings requires reliable uncertainty quantification. Conformal prediction provides a principled framework for constructing prediction sets with finite-sample marginal coverage guarantees (Shafer & Vovk, 2008; Angelopoulos et al., 2024). Applied naively, a common approach is to equip each one-shot predictor with its own conformal prediction set using split conformal calibration. However, data splitting is statistically inefficient in the low-data regimes that motivate one-shot learning (Bashari et al., 2025), and selecting or aggregating predictors adaptively typically violates the assumptions required for conformal validity (Liang et al., 2024).

These challenges motivate the following question: *Can we aggregate the predictions of all one-shot predictors induced by the available labeled data in a manner that adapts to each test input, uses labeled data efficiently for calibration, and still yields valid conformal prediction sets?*

In this work, we answer this question in the affirmative by introducing Conformal Aggregation of One-Shot Predictors (CAOS), a conformal framework for one-shot prediction. At a high level, CAOS adaptively aggregates nonconformity scores across multiple one-shot predictors while allowing all labeled examples to participate in calibration through a leave-one-out construction. Despite breaking classical exchangeability at the score level, CAOS achieves valid finite-sample marginal coverage guarantees.

Our contributions are threefold: (i) we introduce CAOS, a data-efficient conformal framework for aggregating one-shot predictors without requiring disjoint calibration splits; (ii) we establish finite-sample marginal coverage guarantees for CAOS via a novel monotonicity-based reduction to full conformal prediction; and (iii) we demonstrate empiri-

[1]Department of Statistics, University of Wisconsin-Madison, USA. Correspondence to: Maja Waldron <maja.waldron@wisc.edu>.

*Proceedings of the 43rd International Conference on Machine Learning*, Seoul, South Korea. PMLR 306, 2026. Copyright 2026 by the author(s).

cally that CAOS yields substantially smaller prediction sets than split conformal baselines while maintaining reliable coverage in both vision and language one-shot tasks.

## 2. Related Work

In-Context Learning (ICL), first popularized for language models (Brown et al., 2020) and closely related to zero-shot and few-shot prediction paradigms (Bommasani, 2021), is now a standard mechanism for adapting large pretrained models in both language and vision (Radford et al., 2021; Siméoni et al., 2025). In parallel, conformal prediction provides a distribution-free framework for uncertainty quantification with finite-sample marginal coverage guarantees (Shafer & Vovk, 2008; Angelopoulos et al., 2023; 2024), and is particularly appealing in foundation-model settings because it can treat the underlying predictor as a black box.

A growing body of work applies conformal prediction to foundation models in zero-shot, one-shot, and few-shot regimes (Fisch et al., 2021; Quach et al., 2023; Su et al., 2024; Wang et al., 2024; Fillioux et al., 2024); see also (Park et al., 2023; Vishwakarma et al., 2024; Pantelidis et al., 2025; Ye & Wen, 2025; Lin et al., 2025; Tayebati et al., 2025; Silva-Rodríguez et al., 2025a;b; Wang et al., 2025; Xu & Lu, 2025). Many of these methods adopt split conformal calibration for simplicity and modularity. In the one-shot setting, each labeled example induces its own predictor, yielding a *collection* of predictors rather than a single model. Existing conformal approaches typically do not address uncertainty quantification for collections of one-shot predictors whose relevance may vary across test inputs.

The one-shot, low-data regime also sharpens a classical trade-off in conformal prediction between statistical and computational efficiency. Split conformal is computationally efficient but can be statistically inefficient when labeled data are scarce (Bashari et al., 2025), while full conformal can be more statistically efficient but is often computationally prohibitive. Data-reuse schemes such as cross-conformal prediction (Vovk, 2015) and related leave-one-out approaches (Barber et al., 2021) improve data efficiency, but their validity analyses frequently require additional arguments and may introduce slack when the resulting conformity scores are not exchangeable (Gasparin & Ramdas, 2025). These considerations motivate methods that reuse labeled data while retaining sharp finite-sample guarantees.

When confronted with a pool of predictors, a natural strategy is to either *select* a single predictor (Liang et al., 2024; Bai & Jin, 2024; Hegazy et al., 2025) or *aggregate* predictors or their uncertainty sets (Gasparin & Ramdas, 2024b; Rivera et al., 2024; Alami et al., 2025). However, many selection-and aggregation-based conformal approaches preserve validity by relying on disjoint data splits, extra calibration data,

or conservative corrections.

These limitations trace to a central requirement in classical conformal prediction: standard finite-sample guarantees rely on exchangeability of the calibration and test conformity scores (Shafer & Vovk, 2008; Angelopoulos et al., 2023; 2024). When exchangeability is violated at the *data* level, for example under distribution shift (Tibshirani et al., 2019; Barber et al., 2023) or in online prediction (Gasparin & Ramdas, 2024a; Sale & Ramdas, 2025), validity is often recovered via additional assumptions or correction factors that can loosen guarantees (Barber et al., 2023). In the one-shot setting studied here, non-exchangeability instead arises *algorithmically*, because instance-adaptive selection or aggregation over a pool of reference-conditioned predictors breaks exchangeability at the score level, preventing a direct application of classical conformal arguments. We address these challenges with CAOS, which performs instance-adaptive aggregation with leave-one-out calibration while retaining $1 - \alpha$ finite-sample marginal coverage.

## 3. Background

We now formalize the one-shot prediction setting discussed above and introduce the notation used throughout the paper. The basic building blocks of our framework are one-shot predictors and their associated nonconformity scores, while split conformal calibration is reviewed as a reference point.

We assume access to a pool of $n$ labeled examples $\mathcal{D}_n = \{(X_i, Y_i)\}_{i=1}^n$. Each example induces a *one-shot predictor* through a fixed prediction mechanism $\pi$, defined as

$$\pi_i(\cdot \mid x) := \pi(\cdot \mid x; (X_i, Y_i)). \quad (1)$$

Because the predictor $\pi_i$ depends on the specific reference example $(X_i, Y_i)$, predictions for a single test input $x$ may vary substantially across the $n$ reference examples.

To assess how well a candidate label $y$ conforms to a test input $x$ under a one-shot predictor $\pi_i$, we associate each predictor with a *nonconformity score*

$$s_{\pi_i}(x, y) \in \mathbb{R}, \quad (2)$$

where smaller values indicate more conforming predictions. These scores will allow us to convert one-shot predictions into conformal prediction sets. We next describe two instantiations of one-shot prediction and their corresponding nonconformity scores that are used in our experiments.

**Example 1: Landmark Transfer by Patch Similarity.** We consider a landmark localization task in which each reference image is divided into $K$ patches and labeled with $Y_i \in \mathcal{Y} = \{1, \ldots, K\}$ indicating which patch contains the landmark of interest. One-shot landmark transfer leverages the embedding space of a pretrained vision model (e.g.,

DinoV3 (Siméoni et al., 2025)) to predict the location of the landmark in a new image $x$ based on patch similarity. Let $e_x(y)$ denote the embedding of patch $y$ in image $x$.

The nonconformity score induced by reference example $(X_i, Y_i)$ is defined via the softmax over patch similarity, as

$$s_{\pi_i}(x, y) = 1 - \frac{\exp\big(\text{sim}\big(e_x(y), e_{X_i}(Y_i)\big)\big)}{\sum_{y' \in \mathcal{Y}} \exp\big(\text{sim}\big(e_x(y'), e_{X_i}(Y_i)\big)\big)}, \quad (3)$$

where $\text{sim}(\cdot, \cdot)$ denotes cosine similarity. A point prediction corresponds to the most conforming label,

$$\hat{y} = \arg\min_{y \in \mathcal{Y}} s_{\pi_i}(x, y). \quad (4)$$

**Example 2: One-Shot Text Classification with LLMs.**
We next consider text classification with a fixed label set $\mathcal{Y}$ using a Large Language Model (LLM). Given a labeled reference example $(X_i, Y_i)$ and a test input $x$, we construct a prompt of the form

```
Given input Xi, the correct output is Yi.
Now for input x, the output is:     (5)
```

(see Sec. C.3 for the full prompt template), which conditions the model on a single demonstration.

For each candidate label $y \in \mathcal{Y}$, we define the nonconformity score as the average negative log-likelihood of $y$ under the model,

$$s_{\pi_i}(x, y) = \text{AvgNLL}\big(y \mid \text{prompt}(X_i, Y_i, x)\big), \quad (6)$$

where the negative log-likelihood is normalized by the token length of $y$ to avoid penalizing longer labels. As in Eqn. (4), a point prediction is obtained by minimizing this score over $y \in \mathcal{Y}$. While LLMs can condition on multiple in-context examples, in the one-shot setting each reference example induces its own predictor $\pi_i$.

Large pretrained models may yield strong point predictions when used as one-shot predictors. However, there is also a need to assess the reliability of these predictions for a new input. In the next section, we introduce split conformal one-shot prediction, which equips each one-shot predictor $\pi_i$ with a data-dependent prediction set that enjoys finite-sample marginal coverage guarantees.

**Split conformal one-shot prediction.** A natural approach to uncertainty quantification for one-shot predictors is split conformal prediction. Given labeled data split into a reference set $\mathcal{D}_{\text{ref}}$ and a disjoint calibration set $\mathcal{D}_{\text{cal}}$, each reference example $(X_i, Y_i) \in \mathcal{D}_{\text{ref}}$ induces a one-shot predictor $\pi_i$, which can be equipped with a conformal prediction set using calibration data.

For a fixed predictor $\pi_i$, calibration scores are obtained by evaluating the nonconformity score $s_{\pi_i}(X_j, Y_j)$ on calibration examples $(X_j, Y_j) \in \mathcal{D}_{\text{cal}}$, yielding the score set

$$\mathcal{S}^i_{\text{split}} = \{s_{\pi_i}(X_j, Y_j) : (X_j, Y_j) \in \mathcal{D}_{\text{cal}}\}. \quad (7)$$

From this set, a split conformal threshold is computed as

$$\hat{q}^i_{\text{split}} = \text{Quantile}\big(\mathcal{S}^i_{\text{split}} \, ; \, (1 - \alpha)\big(1 + 1/|\mathcal{D}_{\text{cal}}|\big)\big), \quad (8)$$

leading to the prediction set

$$\hat{C}^i_{\text{split}}(X_{n+1}) = \{y \in \mathcal{Y} : s_{\pi_i}(X_{n+1}, y) \leq \hat{q}^i_{\text{split}}\}. \quad (9)$$

Under exchangeability of the calibration data and test point, each $\hat{C}^i_{\text{split}}(X_{n+1})$ satisfies the marginal coverage guarantee

$$\mathbb{P}\big(Y_{n+1} \in \hat{C}^i_{\text{split}}(X_{n+1})\big) \geq 1 - \alpha. \quad (10)$$

Split conformal one-shot prediction yields a valid prediction set for each reference example, but the resulting sets vary widely in size and usefulness, reflecting heterogeneity in one-shot predictor quality. Ideally, predictions would adaptively select or combine the most relevant reference examples for each test input. This creates a basic tension in the one-shot setting: predictive performance benefits from using all labeled examples, while conformal calibration restricts such reuse. We resolve this tension with Conformal Aggregation of One-Shot Predictors (CAOS), a conformal framework that aggregates one-shot predictors while allowing all labeled examples to participate in calibration.

## 4. Method

**Problem setting and goal.** We consider a one-shot prediction setting with $n$ labeled examples $\mathcal{D}_n = \{(X_i, Y_i)\}_{i=1}^n$. Each labeled example induces a one-shot predictor $\pi_i$ with an associated nonconformity score $s_{\pi_i}$. Our goal is to construct prediction sets $\hat{C}_{\text{caos}}(X_{n+1})$ that are small but satisfy finite-sample marginal coverage guarantees. Smaller prediction sets correspond to more informative predictions and reduced predictive uncertainty.

No single reference example is uniformly optimal: the informativeness of one-shot predictors varies substantially across reference examples and may depend on the test input. The central challenge is therefore to design a method that (i) aggregates information across multiple one-shot predictors, (ii) calibrates the resulting aggregate using the same limited labeled data, and (iii) retains finite-sample coverage guarantees. We address this challenge with CAOS, a simple aggregation strategy that thanks to a novel theoretical argument allows all labeled data to participate in aggregation and calibration while maintaining valid coverage.

**CAOS score aggregation.** For notational convenience, given a dataset $\mathcal{D}$ and a target pair $(x, y)$, we define the

---

**Algorithm 1** CAOS Prediction

---

1: **Input:** Labeled data $\mathcal{D}_n = \{(X_i, Y_i)\}_{i=1}^n$; test input $X_{n+1}$; target coverage level $1 - \alpha$; integer $k \geq 1$; one-shot nonconformity score $s_{\pi_i}$.
2: Compute aggregated calibration scores via Eqn. (14):
$S_{\text{caos}}^i \leftarrow s_{\text{caos}}\big((X_i, Y_i)\,;\,\mathcal{D}_n^{-i}\big)$ for $i = 1, \dots, n$.
3: Compute threshold via Eqn. (15):
$\hat{q}_{\text{caos}} \leftarrow \text{Quantile}\big(\{S_{\text{caos}}^i\}_{i=1}^n\,;\,(1-\alpha)(1+1/n)\big)$.
4: Compute test scores for all $y \in \mathcal{Y}$ via Eqn. (13):
$S_{n+1}^y \leftarrow s_{\text{caos}}\big((X_{n+1}, y)\,;\,\mathcal{D}_n\big)$.
5: **Return** calibrated CAOS prediction set via Eqn. (16):
$\hat{C}_{\text{caos}}(X_{n+1}) \leftarrow \{y \in \mathcal{Y} : S_{n+1}^y \leq \hat{q}_{\text{caos}}\}$.

---

multiset of one-shot nonconformity scores

$$\mathcal{A}_{\mathcal{D}}(x, y) := \{s_{\pi_j}(x, y) : (X_j, Y_j) \in \mathcal{D}\}. \qquad (11)$$

That is, $\mathcal{A}_{\mathcal{D}}(x, y)$ collects the nonconformity scores for $(x, y)$ induced by all reference examples in $\mathcal{D}$.

A key observation in the one-shot setting is that only a subset of reference examples are informative for a given test input. For instance, in landmark transfer, reference images with locally similar appearance yield more meaningful patch-similarity scores. Aggregating all $n$ one-shot predictors uniformly therefore dilutes useful information with noise.

To address this, CAOS aggregates scores by focusing on the $k$ *most informative* reference examples. Given a multiset of scores $\mathcal{A} = \{a_i\}_{i=1}^n$ and an integer $k \geq 1$, we define the operator that aggregates the $k$ smallest values in $\mathcal{A}$ as,

$$\Sigma_{\min}^k(\mathcal{A}) := \sum_{j=1}^k a_{(j)}, \qquad (12)$$

where $a_{(1)} \leq a_{(2)} \leq \cdots \leq a_{(n)}$ are the ordered values in $\mathcal{A}$. We will use this operator to select the $k$ reference examples that are most compatible with the candidate output $y$.

For a test input $X_{n+1}$ and candidate output $y$, let $\mathcal{A}_{\mathcal{D}_n}(X_{n+1}, y)$ denote the pool of all one-shot nonconformity scores induced by examples in $\mathcal{D}_n$. The CAOS aggregated nonconformity score is defined as

$$s_{\text{caos}}\big((X_{n+1}, y)\,;\,\mathcal{D}_n\big) := \Sigma_{\min}^k(\mathcal{A}_{\mathcal{D}_n}(X_{n+1}, y)), \quad (13)$$

where $k < n$ is a small constant (we fix $k = 3$; App. D studies sensitivity to $k$). This score is small when several reference examples strongly support the candidate output $y$, and large otherwise. We explicitly indicate the dependence on $\mathcal{D}_n$, since unlike split conformal prediction, CAOS does not maintain disjoint reference and calibration sets.

**CAOS calibration and prediction sets.** To obtain valid prediction sets, the aggregated CAOS nonconformity

scores must be calibrated. We adopt a *leave-one-out* calibration strategy. Specifically, for each labeled example $(X_i, Y_i) \in \mathcal{D}_n$, the CAOS calibration score is computed without using $(X_i, Y_i)$ as a reference example. We therefore define the leave-one-out dataset $\mathcal{D}_n^{-i} = \mathcal{D}_n \setminus (X_i, Y_i)$ and let $\mathcal{A}_{\mathcal{D}_n^{-i}}(X_i, Y_i)$ denote the collection of one-shot nonconformity scores induced by all reference examples in $\mathcal{D}_n^{-i}$.

The CAOS calibration scores are

$$s_{\text{caos}}\big((X_i, Y_i)\,;\,\mathcal{D}_n^{-i}\big) = \Sigma_{\min}^k(\mathcal{A}_{\mathcal{D}_n^{-i}}(X_i, Y_i)). \quad (14)$$

Collecting these scores $S_{\text{caos}}^i := s_{\text{caos}}\big((X_i, Y_i)\,;\,\mathcal{D}_n^{-i}\big)$ over all $i = 1, \dots, n$, we compute the empirical quantile

$$\hat{q}_{\text{caos}} = \text{Quantile}\big(\{S_{\text{caos}}^i\}_{i=1}^n\,;\,(1-\alpha)(1+1/n)\big), \quad (15)$$

which mirrors the finite-sample correction used in split conformal prediction.

The resulting CAOS prediction set for a new input $X_{n+1}$ is

$$\hat{C}_{\text{caos}}(X_{n+1}) = \big\{y \in \mathcal{Y} : s_{\text{caos}}\big((X_{n+1}, y)\,;\,\mathcal{D}_n\big) \leq \hat{q}_{\text{caos}}\big\}. \quad (16)$$

**CAOS' coverage guarantees.** The leave-one-out calibration procedure treats test points differently from calibration data and therefore breaks the classical exchangeability assumptions used in conformal prediction. Yet, we will establish the following result for CAOS.

**Theorem 4.1** (Finite-sample coverage of CAOS)**.** *Assume that $\mathcal{D}_n$ and the test example $(X_{n+1}, Y_{n+1})$ are exchangeable. Then the CAOS prediction set defined in Eqn. (16) satisfies the finite-sample marginal coverage guarantee*

$$\Pr\big(Y_{n+1} \in \hat{C}_{\text{caos}}(X_{n+1})\big) \geq 1 - \alpha. \qquad (17)$$

The proof is nontrivial because the CAOS scores are not exchangeable. In Sec. 5, we establish Thrm. 4.1 via a reduction to a theoretical full conformal variant of CAOS.

## 5. Theory: Coverage Guarantees for CAOS

This section establishes Thrm. 4.1. The proof proceeds via a reduction to a theoretical full conformal version of CAOS. The argument relies on two key observations: (i) a monotonicity property of the CAOS nonconformity score with respect to the reference dataset, and (ii) a set inclusion result showing that CAOS prediction sets contain those produced by full conformal prediction. Together, these imply that CAOS inherits the finite-sample marginal coverage guarantee of full conformal prediction, despite using a computationally cheaper scoring rule that does not yield exchangeable scores.

## 5.1. Theoretical Construction of Full CAOS

Full conformal prediction provides finite-sample marginal coverage but is computationally prohibitive. It requires recalibration for each possible label for each test input. We present a full-conformal version of CAOS not intended for practical use, solely as an analytical tool to establish CAOS' coverage guarantees.

For a test example $X_{n+1}$ and every potential label $y$, we define the augmented dataset $\mathcal{D}_{n+1}^y = \mathcal{D}_n \cup \{(X_{n+1}, y)\}$ and the augmented set of one-shot scores $\mathcal{A}_{\mathcal{D}_{n+1}^y}(X_i, Y_i) = \{s_{\pi_j}(X_i, Y_i) : (X_j, Y_j) \in \mathcal{D}_{n+1}^y\}$. Note that $\mathcal{A}_{\mathcal{D}_{n+1}^y}(X_i, Y_i)$ augments the leave-one-out pool $\mathcal{A}_{\mathcal{D}_n^{-i}}(X_i, Y_i)$ with both the self-score $s_{\pi_i}(X_i, Y_i)$ and the score induced by the hypothetical test pair $(X_{n+1}, y)$. That is, $\mathcal{A}_{\mathcal{D}_{n+1}^y}(X_i, Y_i)$ contains all one-shot nonconformity scores for the target pair $(X_i, Y_i)$ obtained from the augmented set of reference examples. As an alternative to Eqn. (14), we define the full CAOS nonconformity scores

$$\tilde{S}_{\text{full}}^i = \tilde{s}_{\text{caos}}\big((X_i, Y_i); \mathcal{D}_{n+1}^y\big) \qquad (18)$$
$$= \Sigma_{\min}^k\big(\mathcal{A}_{\mathcal{D}_{n+1}^y}(X_i, Y_i) - \{s_{\pi_i}(X_i, Y_i)\}\big),$$

where $-$ denotes the multiset difference as defined in Eqn. (30), App. F. The full CAOS scores differ slightly from the leave-one-out definition of $s_{\text{caos}}$ in Eqn. (14). However, it is deliberately chosen to i) establish exchangeability of full CAOS and ii) such that, the relationship between the scores of CAOS and full CAOS implies that the full conformal coverage guarantees also extend to CAOS.

Based on the calibration scores $\tilde{S}_{\text{full}}^i$, full CAOS requires a separate threshold for each test input $y$

$$\hat{q}_{\text{full}}^y = \text{Quantile}\Big(\{\tilde{S}_{\text{full}}^i\}_{i=1}^n; (1-\alpha)(1+1/n)\Big), \quad (19)$$

to construct the full conformal prediction set

$$\hat{C}_{\text{full}}(X_{n+1}) = \big\{y : \tilde{s}_{\text{caos}}\big((X_{n+1}, y); \mathcal{D}_{n+1}^y\big) \le \hat{q}_{\text{full}}^y\big\}. \tag{20}$$

To establish conformal validity, full conformal prediction requires the scores to be symmetric.

**Lemma 5.1** (Symmetry of $\tilde{s}_{\text{caos}}$). *For any dataset $\mathcal{D}$ and any permutation $\sigma$ of its elements,*

$$\tilde{s}_{\text{caos}}((x, y); \mathcal{D}) = \tilde{s}_{\text{caos}}((x, y); \mathcal{D}_\sigma).$$

*That is, $\tilde{s}_{\text{caos}}$ is symmetric in its dataset argument.*

(Proof in Appendix, Sec. B.1.) By Lemma 5.1 and standard full conformal arguments (Angelopoulos et al., 2024), the prediction set defined in Eqn. (20) achieves finite-sample marginal coverage guarantees at the target level. We prove next that these guarantees extend to CAOS.

## 5.2. Conformal Validity of CAOS

We now prove Thrm. 4.1 by comparing the two constructions of CAOS: the leave-one-out calibrated prediction set defined in Eqn. (16) and the theoretical full conformal construction defined in Eqn. (20). The proof leverages the conformal validity of full CAOS via two steps: (i) a monotonicity lemma showing that the CAOS nonconformity scores can only improve as the reference set grows, and (ii) a comparison lemma relating leave-one-out CAOS scores to their full conformal analogues. Together, these yield the set inclusion $\hat{C}_{\text{full}}(X_{n+1}) \subseteq \hat{C}_{\text{caos}}(X_{n+1})$, which implies coverage.

The following monotonicity lemma formalizes the intuition, that adding more reference examples will only affect the $\Sigma_{\min}^k$ aggregation if it introduces small scores, and in this case, it can only improve the aggregated score.

**Lemma 5.2** (Monotonicity of the CAOS score). *Fix a target pair $(x, y)$ and an integer $k \ge 1$. Let $\mathcal{D} \subseteq \mathcal{D}'$ be two reference datasets with $|\mathcal{D}| \ge k$. Then*

$$s_{\text{caos}}\big((x, y); \mathcal{D}'\big) \le s_{\text{caos}}\big((x, y); \mathcal{D}\big).$$

*That is, the CAOS nonconformity score can only improve (decrease) as the reference set grows.*

Lemma 5.2 (proof in App. B.2) enables a comparison of the CAOS calibration scores with full CAOS.

**Lemma 5.3** (Comparison of CAOS and full CAOS scores). *Given a candidate label $y \in \mathcal{Y}$ and the augmented dataset $\mathcal{D}_{n+1}^y = \mathcal{D}_n \cup \{(X_{n+1}, y)\}$, the test scores are equivalent*

$$\tilde{s}_{\text{caos}}\big((X_{n+1}, y); \mathcal{D}_{n+1}^y\big) = s_{\text{caos}}\big((X_{n+1}, y); \mathcal{D}_n\big), \tag{21}$$
$$\textit{(Test score equivalence)}$$

*while for the calibration scores indexed by $i = 1, \dots, n$,*

$$\tilde{s}_{\text{caos}}\big((X_i, Y_i); \mathcal{D}_{n+1}^y\big) \le s_{\text{caos}}\big((X_i, Y_i); \mathcal{D}_n^{-i}\big), \tag{22}$$
$$\textit{(Calibration score dominance)}.$$

*Proof.* Both statements follow directly from the definitions of the CAOS and full CAOS scores. For the test point $(X_{n+1}, y)$, the full CAOS construction removes the self-score, and the remaining reference set coincides with $D_n$, yielding the test score equivalence. For calibration points, removing the self-score $i$ from the augmented dataset leaves a reference set that strictly contains $D_n^{-i}$, so the inequality in Eqn. (22) follows from the monotonicity of the $\Sigma_k^{\min}$ aggregation (Lemma 5.2). Complete derivations for both cases are provided in Appendix B.3. □

Via a sequence of lemmas, we have established that the nonconformity scores of CAOS and full CAOS match on test inputs, while the CAOS scores dominate full CAOS on calibration inputs. As a result, full CAOS calibration leads

to potentially smaller thresholds, which means that the resulting prediction sets are tighter than the CAOS prediction sets. We formalize this in the next corollary.

**Corollary 5.4** (Set inclusion). *Let $\hat{C}_{\text{full}}(X_{n+1})$ be the full conformal CAOS set defined in Eqn. (20), and let $\hat{C}_{\text{caos}}(X_{n+1})$ be the CAOS set defined in Eqn. (16). Then for any test input $X_{n+1}$,*

$$\hat{\mathcal{C}}_{\text{full}}(X_{n+1}) \subseteq \hat{\mathcal{C}}_{\text{caos}}(X_{n+1}). \tag{23}$$

*Proof.* Let $y \in \hat{C}_{\text{full}}(X_{n+1})$. Using Lemma 5.3, denote $s := \tilde{s}_{\text{caos}}\big((X_{n+1}, y)\,; \mathcal{D}_{n+1}^{y}\big) = s_{\text{caos}}\big((X_{n+1}, y)\,; \mathcal{D}_{n}\big)$. By definition of the prediction set (Eqn. (20)), $s \leq \hat{q}_{\text{full}}^{y}$. By Lemma 5.3, $\tilde{s}_{\text{caos}}\big((X_i, Y_i)\,; \mathcal{D}_{n+1}^{y}\big) \leq s_{\text{caos}}\big((X_i, Y_i)\,; \mathcal{D}_{n}^{-i}\big)$ for all calibration indices. These calibration scores are used to establish $\hat{q}_{\text{caos}}$ and $\hat{q}_{\text{full}}^{y}$ via Eqns. (15) and (19). Since $\text{Quantile}(\cdot\,; (1-\alpha)(1+1/n))$ is monotonic in its first argument, the resulting thresholds obey $\hat{q}_{\text{full}}^{y} \leq \hat{q}_{\text{caos}}$, which means that $s \leq \hat{q}_{\text{caos}}$. Therefore, by Eqn. (16), $y \in \hat{C}_{\text{caos}}(X_{n+1})$. □

*Proof of Thrm. 4.1.* We established $\hat{\mathcal{C}}_{\text{full}}(X_{n+1}) \subseteq \hat{\mathcal{C}}_{\text{caos}}(X_{n+1})$ for any $X_{n+1}$ via Cor. 5.4, so it suffices to show that $\hat{\mathcal{C}}_{\text{full}}$ has $1 - \alpha$ marginal coverage. This follows from standard full conformal arguments, using the symmetry of $\tilde{s}_{\text{caos}}\big((x, y)\,; \mathcal{D}_{n+1}^{y}\big)$ (Lemma 5.1) and the assumed exchangeability of samples from $\mathcal{D}_{n+1}^{y}$. □

**Remarks.** The CAOS coverage guarantee applies to non-conformity scores that satisfy two structural properties: (i) symmetry with respect to permutations of the reference data, and (ii) monotonicity with respect to the inclusion of additional examples. Score functions that violate these conditions, such as scores involving fine-tuning or hyperparameter selection, fall outside the scope of our theory.

Although CAOS deliberately breaks score exchangeability by excluding the test point from the reference data, this does not compromise coverage. The monotonicity property implies that CAOS prediction sets contain those produced by full conformal prediction, rendering score exchangeability unnecessary for marginal coverage guarantees.

## 6. Experiments

We evaluate CAOS as a method for producing calibrated prediction sets for one-shot prediction in comparison to several Split Conformal One-shot Predictor (SCOS) baselines on (i) exemplar-based facial landmarking with pretrained vision models, and (ii) Real World Few-Shot Annotated Tasks (RAFT) with LLMs (Alex et al., 2021).

### 6.1. Exemplar-Based Facial Landmarking

This application is motivated by an ongoing collaboration with pediatric surgeons, who manually annotate patient images pre- and post-operatively to support medical screening and quantitative analysis of surgical outcomes. These analyses rely on task-specific facial landmarks that are not available in standard landmarking datasets and must be defined and annotated by clinical experts.

Clinical annotations are expensive and time-consuming and large-scale annotation is rarely realistic. With a few labeled examples, each annotated image can be used to induce a one-shot predictor. Since relevance varies substantially across predictors, uncertainty-aware adaptive aggregation is essential for reliable deployment. We evaluate CAOS using publicly available face data to enable controlled and reproducible evaluation.

**Facial Landmarking Dataset.** We evaluate exemplar-based facial landmarking using aligned face images from the CelebA dataset. For reproducibility, the exact hash of the dataset version we used along with other relevant implementation details is provided in Appendix C.1.

Each image is divided into $K = 168$ patches of $16 \times 16$ pixels. The label $Y$ for image $X$ indicates in which of the $K$ patches a landmark is located in, i.e. we treat facial landmarking as a classification problem with 168 classes. Other than padding images on the lower right to the next multiple of the patch size, we employ no further preprocessing.

Ground truth landmark locations are obtained via the MediaPipe Face Landmarker with default settings[1], which places 478 landmarks in each image. This auto-annotation scheme provides a scalable way to generate labeled data for 478 landmarking tasks of varying difficulty. While we acknowledge that MediaPipe Face Landmarker could make mistakes, it nevertheless produces a meaningful benchmark for the evaluation of one-shot landmarking via patch similarity.

We randomly select a subset of images, excluding any images where MediaPipe Landmarker fails to place any of the 478 landmarks, to create separate splits $\mathcal{D}_n$ and $\mathcal{D}_{\text{test}}$ with $|\mathcal{D}_n| = |\mathcal{D}_{\text{test}}| = 100$. All conformal approaches are evaluated on the same held-out set $\mathcal{D}_{\text{test}}$. For all split conformal approaches we further split $\mathcal{D}_n$ into a reference set and a calibration set with $|\mathcal{D}_{\text{ref}}| = |\mathcal{D}_{\text{cal}}| = 50$, $\mathcal{D}_{\text{ref}} \cup \mathcal{D}_{\text{cal}} = \mathcal{D}_n$.

**One-Shot Landmark Prediction via Patch Similarity.** Given an example image $X_i$ with a landmark located in patch $Y_i$, the one-shot task is to predict where that same landmark is located in a test image. We instantiate one-shot landmarking using the patch-similarity approach described in Example 1. The embeddings $e_x(y)$ in Eqn. (3), come from a pretrained frozen DINOv3 model (ViT-B/16 pre-

---

[1] `num_faces=1, min_face_det_confidence=0.5`

*Table 1.* Empirical coverage $\widehat{\mathrm{Cov}}$ and prediction set size averaged over landmarks (mean $\pm$ SEM) for exemplar-based facial landmarking, averaged over 478 landmarking tasks, at target miscoverage levels $\alpha \in \{0.05, 0.1, 0.2\}$.

| Method | $\alpha = 0.05$ | | $\alpha = 0.1$ | | $\alpha = 0.2$ | |
| | $\widehat{\mathrm{Cov}}\ (\geq 0.95)$ | $\widehat{\mathrm{Size}}$ | $\widehat{\mathrm{Cov}}\ (\geq 0.9)$ | $\widehat{\mathrm{Size}}$ | $\widehat{\mathrm{Cov}}\ (\geq 0.8)$ | $\widehat{\mathrm{Size}}$ |
| --- | --- | --- | --- | --- | --- | --- |
| SCOS | $0.976 \pm 0.001$ | $36.07 \pm 0.94$ | $0.930 \pm 0.001$ | $21.04 \pm 0.50$ | $0.842 \pm 0.001$ | $13.41 \pm 0.35$ |
| SCOS Best[†] | $0.952 \pm 0.002$ | $20.49 \pm 0.59$ | $0.898 \pm 0.002$ | $12.17 \pm 0.32$ | $0.797 \pm 0.003$ | $7.12 \pm 0.18$ |
| CAOS [ours] | $0.955 \pm 0.001$ | $\mathbf{16.14 \pm 0.58}$ | $0.908 \pm 0.002$ | $\mathbf{9.27 \pm 0.29}$ | $0.810 \pm 0.003$ | $\mathbf{5.29 \pm 0.15}$ |
| SCOS Oracle* | $1.000 \pm 0.000$ | $16.66 \pm 0.58$ | $1.000 \pm 0.000$ | $7.98 \pm 0.27$ | $1.000 \pm 0.000$ | $4.29 \pm 0.13$ |

[†] No marginal coverage guarantee due to model selection on the calibration data; included only as a reference.

* Uses hindsight knowledge of the test label; included as a non-deployable reference.

trained on LVD-1689M) which produces a patch token for each image patch. The register and CLS tokens are dropped.

**Approaches for Conformal One-Shot Landmarking.** We evaluate CAOS in comparison to several split conformal baselines in terms of empirical coverage and prediction set size (Eqns. (28) and (29) in Appendix C.2). Empirical coverage should meet or exceed the target level $1 - \alpha$, while smaller prediction sets, corresponding to lower predictive uncertainty, are preferred. For a fair comparison, all methods use the same one-shot prediction tasks, backbone models, and data splits, differing only in how prediction sets are constructed and calibrated.

- **SCOS:** This baseline follows Alg. 2 to calibrate $|\mathcal{D}_{\mathrm{ref}}|$ one-shot predictors on calibration data $\mathcal{D}_{\mathrm{cal}}$. Performance is reported on average over one-shot predictors.
- **SCOS Best:** This baseline considers the same pool of one-shot predictors and calibration procedure but then uses the calibration set to select the predictor that produces the smallest prediction sets (on average over calibration examples). This selection step, in theory breaks the coverage guarantees associated with split conformal approaches.
- **SCOS Oracle:** This oracle variant is also based on Alg. 2, but leverages access to the test set and the ground-truth label to adaptively select, for each example, the smallest single-source prediction set that contains the true label. While this approach attains perfect coverage, it is not deployable in practice and is included solely as a reference.
- **CAOS [ours]:** CAOS uses $\mathcal{D}_n$ both as the pool of reference examples and as calibration data according to Alg. 1.

**Choice of $k$:** In principle, the aggregation parameter $k$ could be selected via cross-validation. However, in the one-shot setting considered here, labeled data are extremely scarce, and cross-validation would require withholding examples that could otherwise be used for reference or calibration. We therefore fix $k = 3$ for all experiments and empirically verify that CAOS performance is robust to the choice of $k$ across a wide range of values (App. D).

**Results** Table 1 summarizes results in terms of empirical coverage and average set size at different target miscoverage

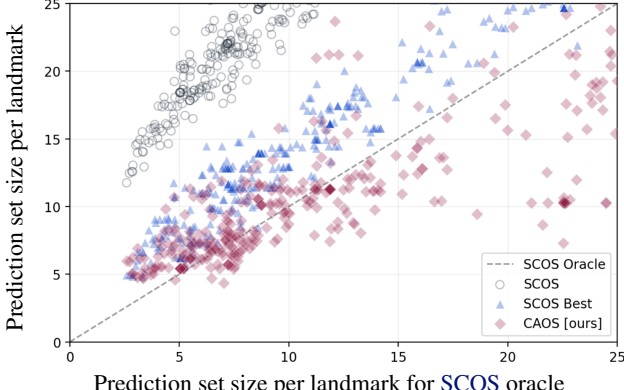

*Figure 1.* Prediction set size per landmark for exemplar-based facial landmarking. Each point corresponds to a landmark. CAOS yields smaller prediction sets than SCOS and SCOS Best, approaching oracle efficiency.

rates $\alpha \in [0.05, 0.1, 0.2]$. We report mean and standard error around the mean averaged over the held-out test data and the 478 landmarking tasks. CAOS consistently meets or exceeds the target coverage rate, while significantly improving over the SCOS prediction sets in terms of size.

The scatter plot in Figure 1 further breaks down these results at $\alpha = 0.05$ by individual landmarks which vary significantly by difficulty; on some landmarks even the oracle that selects the smallest prediction set that contains the true label achieves only prediction set sizes that exceed 20 patches on average (averaged over held-out set). The x-axis is the prediction set size achieved by the SCOS oracle, while the values on the y-axis give the corresponding set size achieved by the other methods, SCOS (gray circles) SCOS Best (blue triangles) and CAOS (red diamonds).

Figure 2 visualizes the one-shot prediction results for 478 facial landmarks in 3 test images. We color each ground truth landmark by the set size achieved by SCOS (top) and CAOS (bottom) (the smaller the better, values below 10 are marked in green). We can see qualitatively that CAOS achieves significantly smaller prediction set sizes.

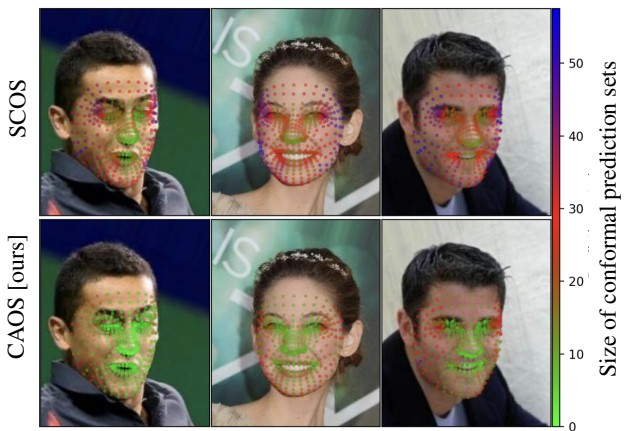

*Figure 2.* Conformal prediction set sizes for one-shot facial landmarking for SCOS (top) and CAOS (bottom) on three test images. Landmarks are colored by prediction set size. Smaller sets (green) correspond to lower predictive uncertainty and are preferable.

To isolate the source of these gains, App. E presents a diagnostic ablation showing that aggregation alone yields limited improvements when data are split, and that reusing labeled examples for both calibration and reference is the main driver for CAOS's efficiency. Thm. 4.1 is essential to establish that such data reuse is valid.

### 6.2. Real-World Few-Shot Annotated Tasks with LLMs

We evaluate CAOS on the RAFT benchmark (Alex et al., 2021), a collection of real-world few-shot text classification tasks designed to reflect practical annotation workflows. Each task provides a small labeled dataset and exhibits substantial heterogeneity in label space size, class balance, and semantic complexity, making RAFT a challenging testbed for uncertainty quantification in low-data regimes.

**One-shot Text Classification with LLMs.** Each RAFT task provides 50 labeled examples. Each example induces a one-shot predictor via ICL with a fixed prompt template (see Sec. C.3), as described in Sec. 3, Ex. 2. For each candidate label $y$ in the task-specific label space, we concatenate the label with the prompt and compute the negative log-likelihood (NLL) of $y$ under the model, and use the length-normalized NLL as the nonconformity score to avoid penalizing longer labels. While we evaluate across Llama-3.2-3B, Qwen3-4B, Llama2-7B, and Qwen3-32B in our experiments, this procedure applies to any language model that provides access to token-level logits, including API-based models, provided the label space is known.

We construct 5 non-overlapping folds, where each fold consists of a dataset $\mathcal{D}_n$ of size 40 and a held-out test set $\mathcal{D}_{\text{test}}$ of size 10, which is used exclusively to evaluate empirical coverage and prediction set size. For split conformal baselines, which require disjoint reference and calibration data,

$\mathcal{D}_n$ is further partitioned into equally sized reference and calibration sets $\mathcal{D}_{\text{ref}}$ and $\mathcal{D}_{\text{calib}}$. All methods use identical data splits, prompts, and backbone model, differing only in how prediction sets are constructed and calibrated.

**Runtime.** For a single predictor (SCOS) calibration and prediction set construction for $m$ test examples requires a computational budget of $O(n + m|\mathcal{Y}|)$. In contrast, CAOS requires evaluating all $n$ possible predictors for each calibration and test example resulting in a computational budget of $O(n^2 + nm|\mathcal{Y}|)$. The additional cost of CAOS arises from evaluating all possible reference predictors whereas SCOS considers only a single reference. The aggregation step introduces an additional factor of $O(\log k)$, but since $k = 3$ is a small constant, we exclude it from the analysis.

In both applications (vision and language) computational cost is dominated by the forward passes of the foundation model. For the landmarking tasks, each image has to be processed by the vision model exactly once to produce the patch level embeddings, so the effective runtimes of CAOS and SCOS are similar to each other with $n+m$ DinoV3 calls. In one-shot prediction with LLMs, each individual score requires a forward pass. On LLaMA-2-7B, the per-fold time average was $49.2s$ for CAOS versus $1.0s$ for SCOS.

**Results.** Table 2 summarizes the performance of CAOS and SCOS on one-shot text classification tasks from the RAFT benchmark at $\alpha = 0.1$ using 4 different LLM backbones. Across language models, CAOS consistently produces smaller prediction sets than SCOS at comparable valid coverage. In addition to average set size and coverage, we report standard deviation over the 9 tasks and 5 folds. As we can see in Figure 3, which shows per-task results for the Llama2-7B model, the high variance in the average set sizes reported in Table 2 comes mostly from task heterogeneity. The middle panel shows the paired difference 95% confidence intervals comparing CAOS per task prediction set size with SCOS. In 4/9 tasks, the prediction sets are comparable while in 5/9 tasks CAOS reduces prediction set size. These results generalize across LLMs and we include complementary figures for the other LLMs in Figures 6 to 9.

## 7. Conclusion

We introduced Conformal Aggregation of One-Shot Predictors (CAOS), a conformal prediction framework designed for one-shot and in-context learning settings where labeled data are scarce and multiple reference-induced predictors naturally arise. CAOS adaptively aggregates nonconformity scores across all available one-shot predictors while allowing every labeled example to participate in calibration.

Despite breaking classical exchangeability assumptions at the score level, we proved that CAOS achieves finite-

| | Llama-3.2-3B | | Qwen3-4B | | Llama-2-7B | | Qwen3-32B | |
|---|---|---|---|---|---|---|---|---|
| | Avg. set size | Emp. cov. | Avg. set size | Emp. cov. | Avg. set size | Emp. cov. | Avg. set size | Emp. cov. |
| SCOS | $1.92 \pm 0.65$ | $0.90 \pm 0.07$ | $1.83 \pm 0.65$ | $0.90 \pm 0.08$ | $1.93 \pm 0.67$ | $0.90 \pm 0.07$ | $1.82 \pm 0.65$ | $0.90 \pm 0.05$ |
| CAOS | $1.80 \pm 0.49$ | $0.91 \pm 0.10$ | $1.64 \pm 0.54$ | $0.92 \pm 0.08$ | $1.70 \pm 0.52$ | $0.90 \pm 0.10$ | $1.70 \pm 0.64$ | $0.90 \pm 0.11$ |

*Table 2.* Comparison of CAOS and SCOS across four LLMs on RAFT one-shot text classification tasks at target coverage $1 - \alpha = 0.9$. We report average prediction set size and empirical coverage (mean $\pm$ standard deviation) across tasks and folds. Across all models, CAOS achieves smaller prediction sets while maintaining valid coverage.

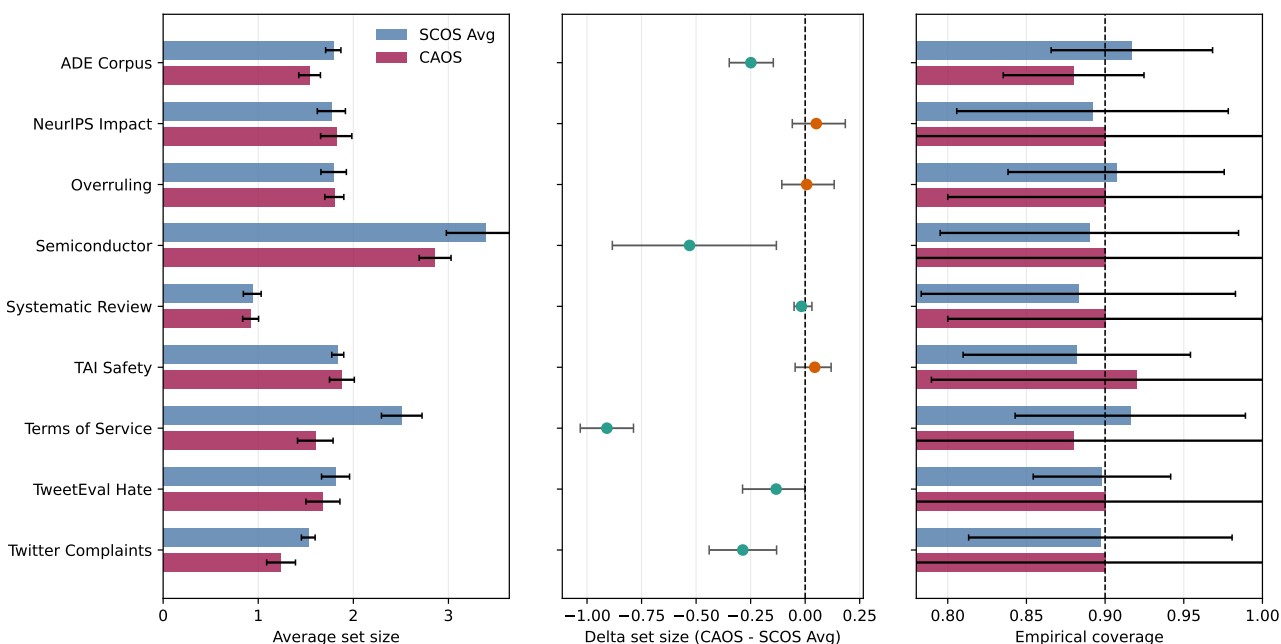

*Figure 3.* RAFT one-shot text classification results for Llama-2-7B at target coverage $1 - \alpha = 0.9$. Left: average prediction set size for CAOS and SCOS. Middle: difference in prediction set size, with negative values indicating smaller, more efficient sets for CAOS. Right: empirical coverage relative to the target coverage level. Across RAFT tasks, CAOS generally produces smaller prediction sets while maintaining comparable coverage.

sample marginal coverage. Our analysis relies on a novel monotonicity-based argument that reduces the CAOS procedure to full conformal prediction, thereby avoiding the slack terms that typically arise in cross-conformal or data-adaptive settings. This result shows that algorithmic departures from exchangeability need not preclude sharp coverage guarantees when carefully structured aggregation rules are used.

Empirically, we demonstrated that CAOS substantially improves efficiency over split conformal baselines in both vision and language applications, yielding significantly smaller prediction sets while maintaining reliable coverage. These gains are especially pronounced in low-data regimes, where data splitting and coarse calibration quantiles severely limit the usefulness of standard conformal approaches.

## Impact Statement

This work develops a methodological contribution to uncertainty quantification in machine learning, with a focus on conformal prediction in one-shot and low-data regimes. The proposed method is intended to improve the statistical efficiency of prediction sets when labeled data are scarce, which is a common setting in scientific and medical applications.

However, as with all uncertainty quantification methods, the established guarantees apply only under the stated assumptions, and misuse or misinterpretation of prediction sets could lead to unwarranted confidence. The method does not introduce new data collection, deployment, or automation mechanisms, and we do not anticipate specific negative societal impacts beyond those generally associated with the application domains in which conformal prediction is used.

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

## A. Pseudocode

---

**Algorithm 2** Split Conformal One-Shot Prediction

---

1: **Input:** Example $(X_i, Y_i)$ defining one-shot predictor $i$; calibration set $\mathcal{D}_{\text{cal}}$ of size $|\mathcal{D}_{\text{cal}}|$; test point $X_{n+1}$; target coverage level $1 - \alpha$; nonconformity score $s_{\pi_i}$.

2: Compute set of calibration scores via Eqn. (7):
$$\mathcal{S}^i_{\text{split}} \leftarrow \left\{ s_{\pi_i}(X_j, Y_j) : (X_j, Y_j) \in \mathcal{D}_{\text{cal}} \right\}.$$

3: Compute threshold via Eqn. (8):
$$\hat{q}^i_{\text{split}} \leftarrow \text{Quantile}\left( \mathcal{S}^i_{\text{split}} \,;\, (1 - \alpha)\left(1 + 1/|\mathcal{D}_{\text{cal}}|\right) \right).$$

4: **Return** prediction set via Eqn. (9):
$$\hat{C}^i_{\text{split}}(X_{n+1}) \leftarrow \left\{ y \in \mathcal{Y} : s_{\pi_i}(X_{n+1}, y) \leq \hat{q}^i_{\text{split}} \right\}.$$

---

**Algorithm 3** CAOS Prediction (repeat of Alg. 1)

---

1: **Input:** Labeled data $\mathcal{D}_n = \{(X_i, Y_i)\}^n_{i=1}$; test input $X_{n+1}$; target coverage level $1 - \alpha$; integer $k \geq 1$; one-shot nonconformity score $s_{\pi_i}$.

2: Compute aggregated calibration scores via Eqn. (14):
$$S^i_{\text{caos}} \leftarrow s_{\text{caos}}\left( (X_i, Y_i) \,;\, \mathcal{D}_n^{-i} \right) \text{ for } i = 1, \dots, n.$$

3: Compute threshold via Eqn. (15):
$$\hat{q}_{\text{caos}} \leftarrow \text{Quantile}\left( \{S^i_{\text{caos}}\}^n_{i=1} \,;\, (1 - \alpha)(1 + 1/n) \right).$$

4: Compute test scores for all $y \in \mathcal{Y}$ via Eqn. (13):
$$S^y_{n+1} \leftarrow s_{\text{caos}}\left( (X_{n+1}, y) \,;\, \mathcal{D}_n \right).$$

5: **Return** calibrated CAOS prediction set via Eqn. (16):
$$\hat{C}_{\text{caos}}(X_{n+1}) \leftarrow \{y \in \mathcal{Y} : S^y_{n+1} \leq \hat{q}_{\text{caos}}\}.$$

---

**Algorithm 4** Full CAOS Prediction (Theoretical)

---

1: **Input:** Labeled data $\mathcal{D}_n = \{(X_i, Y_i)\}^n_{i=1}$; test input $X_{n+1}$; target coverage level $1 - \alpha$; integer $k \geq 1$; one-shot nonconformity score $s_{\pi.}(\cdot,)$.

2: **for** each $y \in \mathcal{Y}$ **do**

3:     Compute full CAOS calibration scores via Eqn. (18):
$$\tilde{S}^i_{\text{full}} \leftarrow \tilde{s}_{\text{caos}}\left( (X_i, Y_i) \,;\, \mathcal{D}^y_{n+1} \right) \text{ for } i = 1, \dots, n.$$

4:     Compute test score via Eqns. (13) and (21):
$$S^y_{n+1} \leftarrow \tilde{s}_{\text{caos}}\left( (X_{n+1}, y) \,;\, \mathcal{D}^y_{n+1} \right)$$

5:     Compute full conformal threshold via Eqn. (19):
$$\hat{q}^y_{\text{full}} \leftarrow \text{Quantile}\left( \{\tilde{S}^i_{\text{full}}\}^n_{i=1} \,;\, (1 - \alpha)(1 + 1/n) \right).$$

6: **end for**

7: **Return** full CAOS prediction set via Eqn. (20):
$$\hat{\mathcal{C}}_{\text{full}}(X_{n+1}) \leftarrow \left\{ y \in \mathcal{Y} : S^y_{n+1} \leq \hat{q}^y_{\text{full}} \right\}.$$

---

## B. Detailed Proofs for CAOS Theory

### B.1. Proof of Symmetry of the full CAOS score

*Proof of Lemma 5.1.* Fix $(x, y)$ and a dataset $\mathcal{D}$. Let $\mathcal{A}_{\mathcal{D}}(x, y)$ denote the multiset of one-shot nonconformity scores obtained by using each element of $\mathcal{D}$ as a reference example for the target $(x, y)$. Permuting the elements of $\mathcal{D}$ does not change this multiset, hence $\mathcal{A}_{\mathcal{D}}(x, y) = \mathcal{A}_{\mathcal{D}_\sigma}(x, y)$.

The operator $\Sigma_{\min}^k$ depends only on the multiset of its inputs and is therefore symmetric. Moreover, the subtracted term in Eqn. (18) corresponds to the one-shot score induced by the target example itself and is unaffected by permutations of the remaining elements of $\mathcal{D}$. It follows that $\tilde{s}_{\text{caos}}((x, y); \mathcal{D}) = \tilde{s}_{\text{caos}}((x, y); \mathcal{D}_\sigma)$. □

### B.2. Proof of the Monotonicity of the CAOS score

*Proof of Lemma 5.2.* Let $\mathcal{A} = \mathcal{A}_{\mathcal{D}}(x, y)$ and $\mathcal{A}' = \mathcal{A}_{\mathcal{D}'}(x, y)$. Since $\mathcal{D} \subseteq \mathcal{D}'$, the multiset $\mathcal{A}'$ contains all scores in $\mathcal{A}$.

Let $a_{(1)} \leq \cdots \leq a_{(|\mathcal{A}|)}$ denote the ordered values of $\mathcal{A}$ and let $a'_{(1)} \leq \cdots \leq a'_{(|\mathcal{A}'|)}$ denote the ordered values of $\mathcal{A}'$. For each $j \leq k$ we have $a'_{(j)} \leq a_{(j)}$, because adding elements to a multiset cannot increase any of the first $k$ order statistics. Therefore,

$$\Sigma_{\min}^k(\mathcal{A}') = \sum_{j=1}^k a'_{(j)} \leq \sum_{j=1}^k a_{(j)} = \Sigma_{\min}^k(\mathcal{A}),$$

yielding $s_{\text{caos}}((x, y); \mathcal{D}') \leq s_{\text{caos}}((x, y); \mathcal{D})$, as claimed. □

### B.3. Score comparison between CAOS and full CAOS

Fix a candidate label $y \in \mathcal{Y}$ and the augmented dataset $\mathcal{D}_{n+1}^y = \mathcal{D}_n \cup \{(X_{n+1}, y)\}$. Our goal is to show that the test scores are equivalent

$$\tilde{s}_{\text{caos}}\big((X_{n+1}, y); \mathcal{D}_{n+1}^y\big) = s_{\text{caos}}\big((X_{n+1}, y); \mathcal{D}_n\big), \tag{24}$$

while for the calibration scores indexed by $i = 1, \ldots, n$,

$$\tilde{s}_{\text{caos}}\big((X_i, Y_i); \mathcal{D}_{n+1}^y\big) \leq s_{\text{caos}}\big((X_i, Y_i); \mathcal{D}_n^{-i}\big). \tag{25}$$

*Proof of Lemma 5.3.* Both statements follow from the definitions of the scores and the definition of multiset subtraction provided in Eqn. (30), App. F.

For the test inputs we have

$$\begin{aligned}
\tilde{s}_{\text{caos}}\big((X_{n+1}, y); \mathcal{D}_{n+1}^y\big) &= \Sigma_{\min}^k\big(\mathcal{A}_{\mathcal{D}_{n+1}^y}(X_{n+1}, y) - \{s_{\pi_{n+1}^y}(X_{n+1}, y)\}\big) \\
&= \Sigma_{\min}^k\big(\mathcal{A}_{\mathcal{D}_n}(X_{n+1}, y)\big) \\
&= s_{\text{caos}}\big((X_{n+1}, y); \mathcal{D}_n\big).
\end{aligned} \tag{26}$$

Similarly, for the calibration scores:

$$\begin{aligned}
\tilde{s}_{\text{caos}}\big((X_i, Y_i); \mathcal{D}_{n+1}^y\big) &= \Sigma_{\min}^k\big(\mathcal{A}_{\mathcal{D}_{n+1}^y}(X_i, Y_i) - \{s_{\pi_i}(X_i, Y_i)\}\big) \\
&= \Sigma_{\min}^k\big(\mathcal{A}_{\mathcal{D}_n^{-i} \cup \{(X_{n+1}, y)\}}(X_i, Y_i)\big) \\
&= s_{\text{caos}}\big((X_i, Y_i); \mathcal{D}_n^{-i} \cup \{(X_{n+1}, y)\}\big) \\
&\leq s_{\text{caos}}\big((X_i, Y_i); \mathcal{D}_n^{-i}\big).
\end{aligned} \tag{27}$$

The last inequality follows from Lemma 5.2 (Monotonicity Lemma). □

# C. Additional Implementation Details

### C.1. Facial Landmarking Data

We use the Hugging Face dataset nielsr/CelebA-faces (aligned face crops of CelebA), train split (202,599 images; revision db26feecbaed40316ee4c80ec0e72211a2f4afd6), with original resolution 178×218 and no additional resizing.

### C.2. Evaluation Metrics

All conformal prediction methods are evaluated in terms of empirical coverage and average prediction set size.

**Coverage.** Marginal coverage reports the fraction of prediction sets that contain the true label for labeled examples in the test set $\mathcal{D}_{\text{test}}$. We estimate empirical coverage as

$$\widehat{\text{Cov}} = \frac{1}{|\mathcal{D}_{\text{test}}|} \sum_{(X,Y) \in \mathcal{D}_{\text{test}}} \mathbb{1}\{Y \in \hat{\mathcal{C}}(X)\}, \tag{28}$$

reported either averaged per-task, or averaged jointly over all tasks. Even though Split Conformal One-shot Predictor (SCOS) and CAOS enjoy finite-sample marginal coverage guarantees, it can still occur that individual results lie below the target coverage of $1 - \alpha$. This is particularly prone to happen for small test sets $\mathcal{D}_{\text{test}}$.

**Set size.** We report the average prediction set size

$$\widehat{\text{Size}} = \frac{1}{|\mathcal{D}_{\text{test}}|} \sum_{(X,Y) \in \mathcal{D}_{\text{test}}} |\hat{\mathcal{C}}(X)|. \tag{29}$$

For the same marginal coverage, smaller sets are preferable, as they correspond to lower predictive uncertainty.

### C.3. LLM Prompt Template.

```
Prompt template

This is a text classification task.
You will get one example, then predict the most appropriate label.
Return only the label.

Example:
Text: <source_text>
Label: <source_label>

Text: <target_text>
Label:
```

# D. Robustness to the Aggregation Parameter $k$

**Setup.** All landmarking experiments in the main paper use a fixed aggregation parameter $k = 3$. To assess sensitivity to this choice, we evaluate CAOS across a range of values $k \in \{1, \ldots, 10\}$ on exemplar-based facial landmarking. For each landmarking task, we report prediction set size averaged over the evaluation set, using the same calibration and evaluation splits as in the main experiments.

**Average robustness.** Figure 4 (left) shows the average prediction set size across all landmarking tasks as a function of $k$. Across a wide range of aggregation levels, average performance remains essentially constant, indicating that CAOS does not require careful tuning of $k$ to achieve reliable efficiency.

**Task-level heterogeneity.** Figure 4 (right) shows prediction set size for individual landmarking tasks. We observe heterogeneous behavior: for many tasks, averaged over test examples, performance is stable w.r.t. $k$, but for some landmarks,

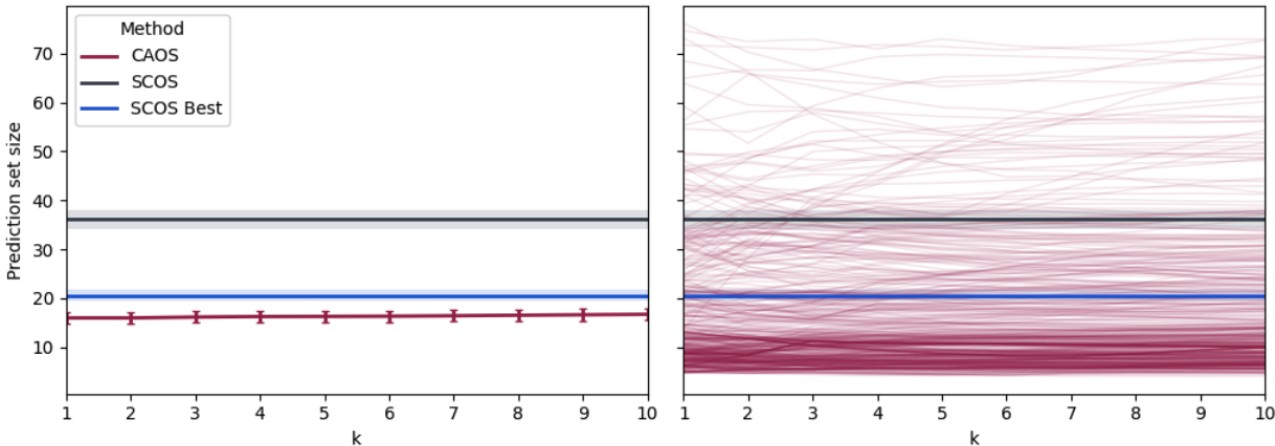

*Figure 4.* Sensitivity of CAOS to the aggregation parameter $k$ on exemplar-based facial landmarking. **Left:** Average prediction set size across all landmarking tasks as a function of $k$. **Right:** Per-landmark prediction set size for CAOS (spaghetti plot). While individual landmarks exhibit heterogeneous responses to increasing $k$, these effects balance out in aggregate, resulting in stable average performance across a wide range of $k$.

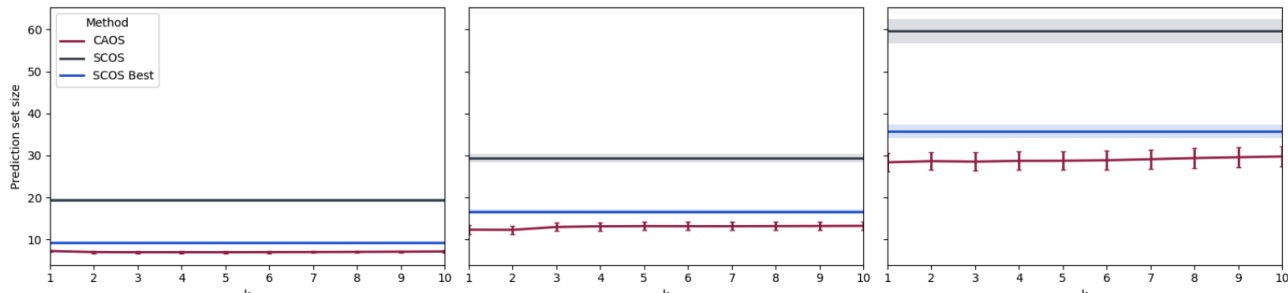

*Figure 5.* CAOS sensitivity to the aggregation parameter $k$, stratified by landmarking task difficulty. Landmarks are grouped into easy, medium, and hard regimes based on oracle prediction set size. Within each difficulty regime, average prediction set size remains stable across $k$, indicating that robustness to $k$ is not driven by a particular subset of tasks.

increasing $k$ leads to improved efficiency, while for others it results in larger prediction sets. These opposing trends largely cancel out at the population level, explaining the observed stability of the average performance.

**Stratification by task difficulty.** To further verify that robustness to $k$ holds across different regimes, we stratify landmarks by oracle prediction set size into easy, medium, and hard tasks. As shown in Figure 5, CAOS exhibits stable performance with respect to $k$ within each difficulty regime. This confirms that robustness to the aggregation parameter is not driven solely by easy landmarking tasks.

**Conclusion.** Together, these results demonstrate that CAOS is robust to the choice of aggregation parameter $k$. While the optimal value of $k$ may vary across individual tasks, overall performance remains stable without tuning, supporting the use of a fixed, task-agnostic aggregation level in practice.

## E. Data-Reuse Ablation Study

A key feature of CAOS is its ability to reuse the same labeled examples for both calibration and reference, while still maintaining finite-sample marginal coverage at level $1 - \alpha$. To disentangle the contributions of aggregation and data reuse, we evaluate several CAOS variants that selectively restrict how labeled data are used for calibration and reference. Specifically, we consider variants that follow the split conformal data splitting protocol of SCOS and split the data into disjoint calibration and reference pools. To better disentangle data scarcity in the reference pool vs. the calibration data, we also study variants that reuse data for only one of these roles. Table 3 reports average prediction set size and empirical

*Table 3.* Prediction set size and empirical coverage for CAOS variants and baselines on exemplar-based facial landmarking at $\alpha = 0.05$. Results are averaged over landmarks. Here, $\mathcal{D}_n$ denotes the full labeled dataset, while $\mathcal{D}_{\text{cal}}$ and $\mathcal{D}_{\text{ref}}$ are disjoint calibration and reference splits with $|\mathcal{D}_{\text{cal}}| = |\mathcal{D}_{\text{ref}}| = |\mathcal{D}_n|/2$.

| Method | Calibration Data | Reference Pool | Avg. Set Size | Empirical Coverage |
|---|---|---|---|---|
| SCOS | $\mathcal{D}_{\text{cal}}$ | $\mathcal{D}_{\text{ref}}$ | 36.07 | 0.976 |
| Split CAOS (ref + cal) | $\mathcal{D}_{\text{cal}}$ | $\mathcal{D}_{\text{ref}}$ | 33.54 | 0.955 |
| Split CAOS (cal) | $\mathcal{D}_{\text{cal}}$ | $\mathcal{D}_n$ | 31.91 | 0.980 |
| Split CAOS (ref) | $\mathcal{D}_n$ | $\mathcal{D}_{\text{ref}}$ | 17.47 | 0.954 |
| CAOS [ours] | $\mathcal{D}_n$ | $\mathcal{D}_n$ | **16.14** | 0.955 |

coverage for each variant.

The results in Table 3 show that aggregation alone provides limited gains when data are split, while joint reuse of labeled data for both calibration and reference is essential for achieving the efficiency improvements observed with CAOS.

## F. Notation Summary

This appendix summarizes the notation used throughout the paper.

**Data and indices.**

- $\mathcal{X}$: input space.
- $\mathcal{Y}$: output (label) space.
- $\mathcal{D}_n = \{(X_i, Y_i)\}_{i=1}^n$: labeled reference dataset.
- $(X_{n+1}, Y_{n+1})$: test example with ground truth label $Y_{n+1}$.
- $\mathcal{D}_n^{-i} = \mathcal{D}_n \setminus \{(X_i, Y_i)\}$: leave-one-out dataset for index $i$.
- $\mathcal{D}_{n+1}^y = \mathcal{D}_n \cup \{(X_{n+1}, y)\}$: dataset augmented with a hypothetical test label $y$.
- $i, j$: indices of reference examples.
- $y \in \mathcal{Y}$: candidate output label.

**One-shot predictors and nonconformity scores.**

- $\pi_i(\cdot \mid x)$: one-shot predictor induced by reference example $(X_i, Y_i)$.
- $s_{\pi_i}(x, y)$: one-shot nonconformity score for target $(x, y)$ using predictor $\pi_i$.
- $\mathcal{A}_{\mathcal{D}}(x, y) = \{s_{\pi_j}(x, y) : (X_j, Y_j) \in \mathcal{D}\}$: pool of one-shot nonconformity scores for target $(x, y)$ based on one-shot predictors induced by examples in $\mathcal{D}$. All score pools are interpreted as multisets.

**Aggregation and Quantile operators.**

- $\Sigma_{\min}^k(\mathcal{A})$: sum of the $k$ smallest elements of multiset $\mathcal{A}$.
- $\text{Quantile}(\mathcal{S} ; \tau) := s_{(\min\{\lceil \tau(|\mathcal{S}|)\rceil, |\mathcal{S}|\})}$, where $s_{(1)} \le \cdots \le s_{(|\mathcal{S}|)}$ are the order statistics of $\mathcal{S}$.
- $\mathcal{A} - \{x\}$: multiset difference obtained from $\mathcal{A}$ by removing one occurrence of $x$ (if present). Equivalently,

$$m_{\mathcal{A}-\{x\}}(y) := \begin{cases} \max(m_{\mathcal{A}}(y) - 1, 0), & \text{if } y = x, \\ m_{\mathcal{A}}(y), & \text{otherwise,} \end{cases} \tag{30}$$

where $m_{\mathcal{A}}(y)$ denotes the multiplicity of $y$ in the multiset $\mathcal{A}$.

- The aggregation and quantile operators depend only on the multiset of values and are permutation invariant.

**Split conformal, CAOS, and full CAOS nonconformity scores.**

- Test-time scores
  - $s_{\pi_i}(X_{n+1}, y)$: split conformal one-shot nonconformity score induced by reference example $i$.
  - $s_{\pi_{n+1}^y}(X_{n+1}, y)$: self-score of example $(X_{n+1}, y)$ used to predict itself.
  - $s_{\mathrm{caos}}\big((X_{n+1}, y)\,;\,\mathcal{D}_n\big) = \Sigma_{\min}^k(\mathcal{A}_{\mathcal{D}_n}(X_{n+1}, y))$: CAOS test-time aggregated nonconformity score.
  - $\tilde{s}_{\mathrm{caos}}\big((X_{n+1}, y)\,;\,\mathcal{D}_{n+1}^y\big) = \Sigma_{\min}^k\big(\mathcal{A}_{\mathcal{D}_{n+1}^y}(X_{n+1}, y) - \{s_{\pi_{n+1}^y}(X_{n+1}, y)\}\big)$: full CAOS test-time nonconformity score. The multiset difference $-$ is defined in Eqn. (30).

- Calibration scores
  - $s_{\pi_i}(X_j, Y_j)$: split conformal calibration score for example $j$ based on one-shot predictor $\pi_i$.
  - $s_{\mathrm{caos}}\big((X_i, Y_i)\,;\,\mathcal{D}_n^{-i}\big) = \Sigma_{\min}^k(\mathcal{A}_{\mathcal{D}_n^{-i}}(X_i, Y_i))$: leave-one-out CAOS calibration score for example $i$.
  - $\tilde{s}_{\mathrm{caos}}\big((X_i, Y_i)\,;\,\mathcal{D}_{n+1}^y\big) = \Sigma_{\min}^k\big(\mathcal{A}_{\mathcal{D}_{n+1}^y}(X_i, Y_i) - \{s_{\pi_i}(X_i, Y_i)\}\big)$: full CAOS calibration score for example $i$.

- Relationships between nonconformity scores
  - $\tilde{s}_{\mathrm{caos}}\big((X_{n+1}, y)\,;\,\mathcal{D}_{n+1}^y\big) = s_{\mathrm{caos}}\big((X_{n+1}, y)\,;\,\mathcal{D}_n\big)$: test-time score equivalence between CAOS and full CAOS.
  - $\tilde{s}_{\mathrm{caos}}\big((X_i, Y_i)\,;\,\mathcal{D}_{n+1}^y\big) \le s_{\mathrm{caos}}\big((X_i, Y_i)\,;\,\mathcal{D}_n^{-i}\big)$: full CAOS calibration scores dominate CAOS.

**Quantiles and prediction sets.**

- $\hat{q}_{\mathrm{split}}^i$: split conformal threshold for one-shot predictor $\pi_i$.

- $\hat{q}_{\mathrm{caos}}$: CAOS calibration threshold.

- $\hat{q}_{\mathrm{full}}^y$: full CAOS calibration threshold.

- $\hat{C}_{\mathrm{split}}^i(X_{n+1})$: split conformal one-shot prediction set for predictor $\pi_i$

- $\hat{C}_{\mathrm{caos}}(X_{n+1})$: CAOS prediction set.

- $\hat{C}_{\mathrm{full}}(X_{n+1})$: full conformal CAOS prediction set.

# G. Additional results on RAFT

To complement the RAFT results in the main paper, we provide additional results that show that CAOS' performance generalizes accross Large Language Models (LLMs).

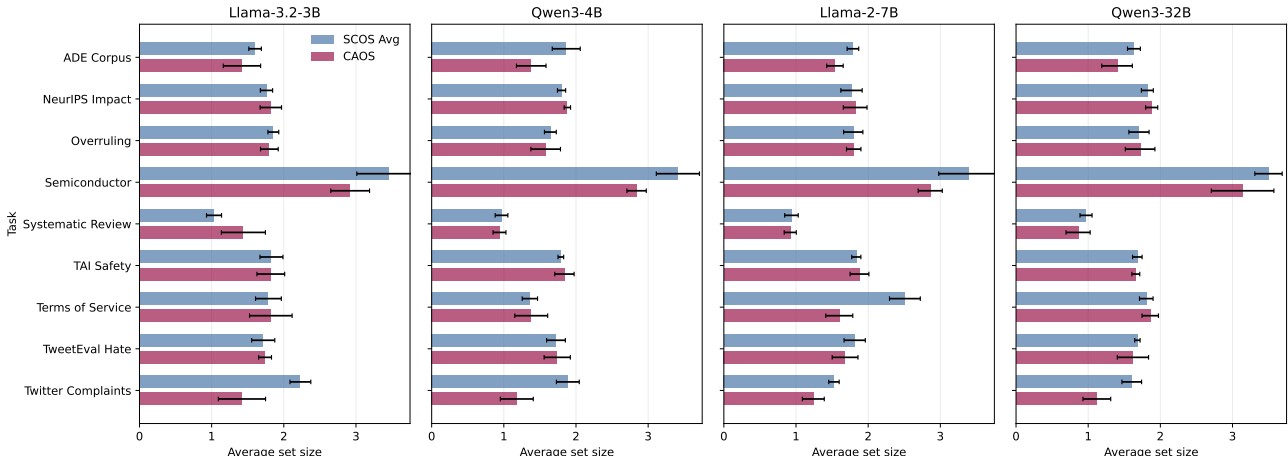

*Figure 6.* Average prediction set size on RAFT one-shot text classification tasks at target coverage $1 - \alpha = 0.9$, shown across all four LLMs. Across models and tasks, CAOS generally produces smaller prediction sets than SCOS.

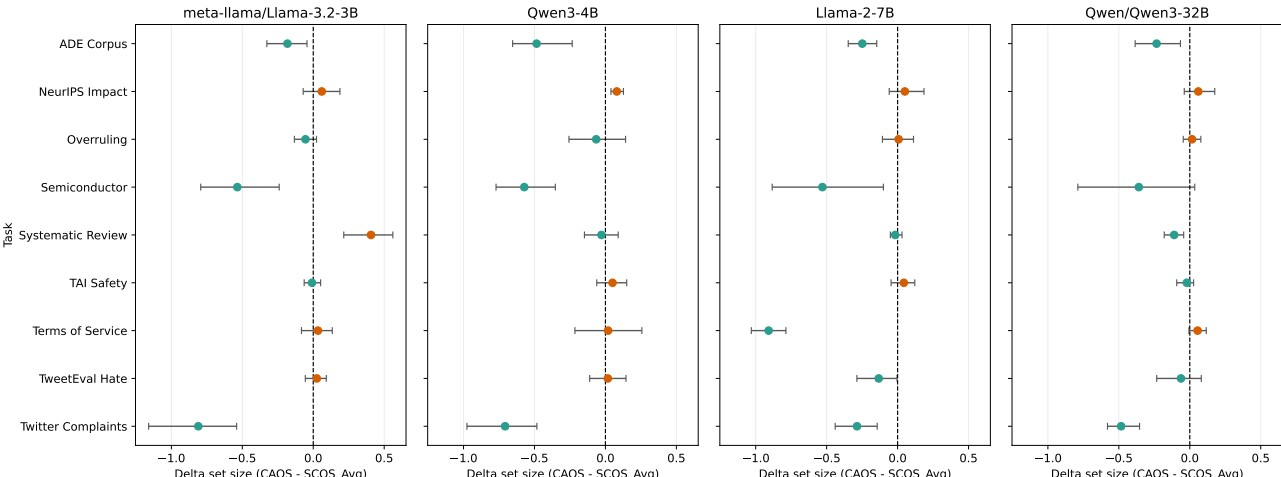

*Figure 7.* Difference in average prediction set size between CAOS and SCOS on RAFT one-shot text classification tasks across all four LLMs. Negative values indicate smaller prediction sets for CAOS.

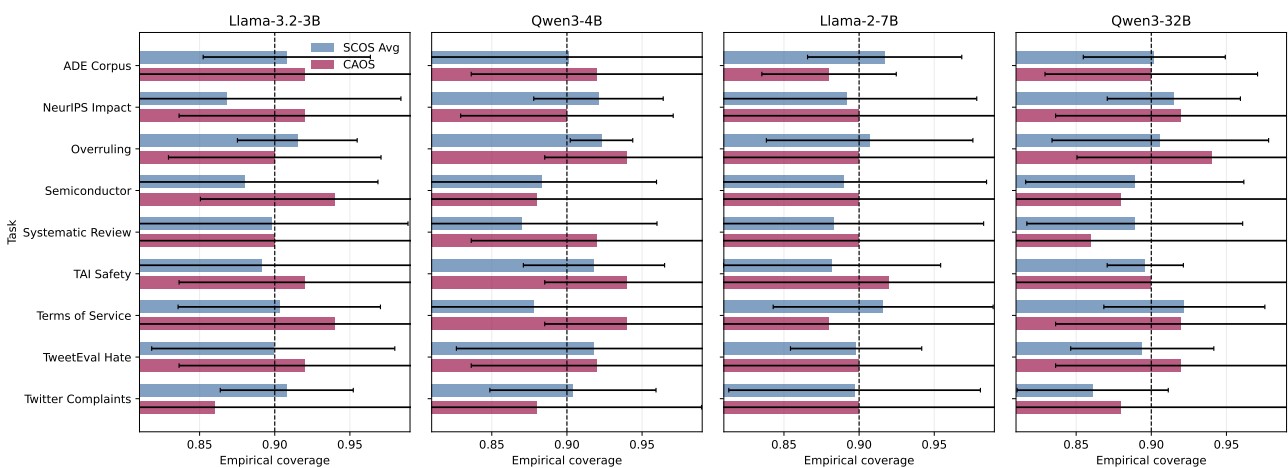

*Figure 8.* Empirical coverage on RAFT one-shot text classification tasks at target coverage $1 - \alpha = 0.9$, shown across all four LLMs. CAOS maintains coverage comparable to SCOS while improving prediction set efficiency.

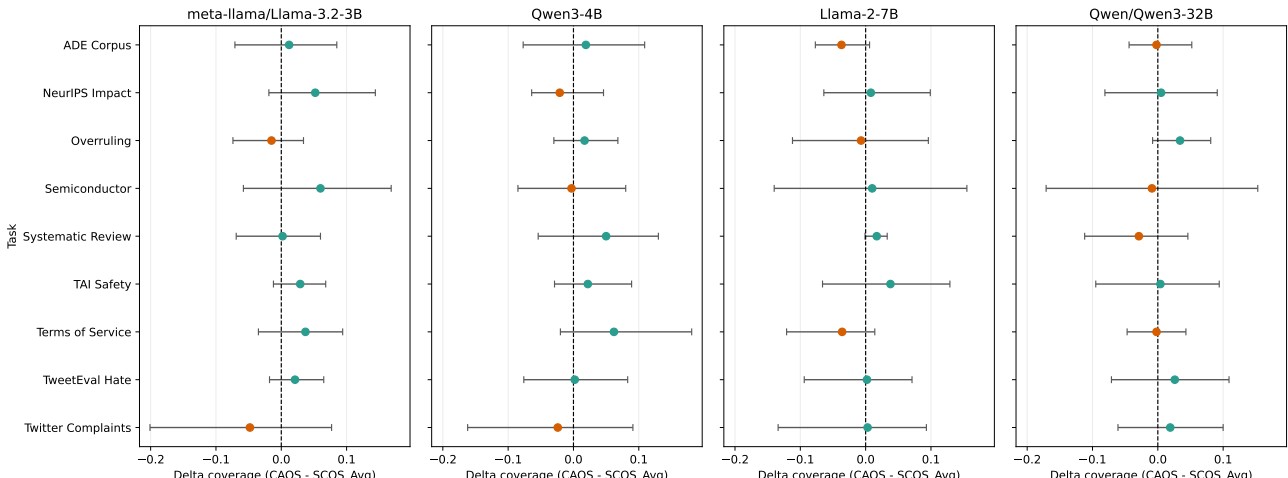

*Figure 9.* Difference in empirical coverage between CAOS and SCOS on RAFT one-shot text classification tasks across all four LLMs. Values near zero indicate comparable coverage, while positive values indicate higher empirical coverage for CAOS.

