# OpenReview forum: "CAOS: Conformal Aggregation of One-Shot Predictors"
_ICML.cc/2026/Conference — ICML 2026 regular_

### Official Review · Reviewer_76Fx · 2026-03-11

**Soundness:** 3
**Presentation:** 3
**Significance:** 3
**Originality:** 3
**Overall Recommendation:** 5
**Confidence:** 3

**Summary:**

The paper investigates conformal prediction when predictions are produced by aggregating multiple one-shot predictors derived from labeled examples. The authors show that instance-adaptive aggregation breaks the score exchangeability required by standard conformal prediction and thus invalidates its finite-sample coverage guarantees. To address this issue, they propose a leave-one-out symmetrized scoring construction that restores exchangeability while retaining adaptive predictor aggregation. Theoretical results establish finite-sample marginal coverage, and experiments demonstrate the method on vision and language tasks.

**Compliance With Llm Reviewing Policy:**

Affirmed.

**Final Justification:**

The authors successfully addressed my concerns regarding baselines and connections to prior work. Their rebuttal clarified that CAOS excels in data-scarce regimes where traditional ensemble methods struggle due to the dual use of data for both model induction and calibration. Specifically, the added comparison to Jackknife+ demonstrates that CAOS provides tighter, more efficient prediction sets while maintaining valid $1−\alpha$ coverage through its principled monotonicity-based argument. Given the theoretical clarity and its high relevance to retrieval-based prediction, I recommend the paper for acceptance.

**Key Questions For Authors:**

1. Could the authors clarify why the experiments do not compare against methods that aggregate multiple predictors (e.g., ensemble-style conformal approaches)?
2. The paper mentions data-reuse approaches such as cross-conformal prediction and leave-one-out schemes (e.g., Barber et al., 2021). I’m curious why jackknife+ or similar methods are not discussed further or used as baselines, since they also rely on leave-one-out constructions.

**Limitations:**

yes

**Strengths And Weaknesses:**

**Strengths**:
1. Studies an interesting setting of conformal prediction with multiple one-shot predictors, which is relevant for retrieval- or demonstration-based prediction scenarios.
2. The paper analyzes how adaptive predictor aggregation affects the exchangeability assumption underlying standard conformal prediction.
3. The proposed leave-one-out style construction provides a principled way to address this issue, and the paper presents theoretical analysis supporting the finite-sample marginal coverage guarantee.
4. The overall proof structure is reasonably clear and the theoretical argument is relatively easy to follow.

**Weakness**:
1. The set of experimental baselines is limited, mostly variants of split conformal on individual predictors.
2. The relationship to existing leave-one-out / jackknife-style conformal methods could be clarified further.

---

> ### Author Rebuttal · Authors · 2026-03-28
>
> We thank the reviewer for the thoughtful feedback and for recognizing the relevance of the setting and the clarity of the theoretical analysis.
>
> ### **Aggregation baselines**
>
> We agree that aggregation-based conformal baselines are important to consider. However, most existing ensemble-style conformal approaches assume a collection of predictors that have already been trained independently, together with an additional labeled dataset used to learn aggregation weights and/or perform conformal calibration, typically under a split-conformal protocol.
>
> In contrast, in our setting each predictor is induced from the same scarce labeled dataset that must also support calibration. This creates a dependence structure and data-efficiency constraint that standard conformal ensemble methods do not directly address. Our goal in this work is to study whether valid and efficient uncertainty quantification is possible in this regime while aggressively reusing data. The comparison to SCOS isolates the benefit of this reuse, and Table 2 in the appendix further suggests that a substantial portion of the efficiency gain arises from data reuse itself. Developing richer aggregation strategies for this regime, while preserving the required theoretical structure, is an interesting direction for future work, and one we are actively pursuing.
>
> ### **Relation to jackknife+ and leave-one-out conformal methods**
>
> We thank the reviewer for raising this connection. CAOS is indeed closely related in spirit to jackknife+-style methods: both are data-reuse approaches that use leave-one-out constructions, and both generate predictions for calibration point i using the dataset with that point removed. In jackknife+, this corresponds to retraining a predictor on $D_n^{-i}$ while in CAOS it corresponds to aggregating predictions from reference-conditioned predictors induced by $D_n^{-i}$.
>
> The key difference lies in how prediction sets are constructed and the resulting coverage guarantees. If one were to apply a standard jackknife+-style calibration and prediction rule to the CAOS aggregated nonconformity scores, the resulting guarantee would be $1 - 2 \alpha$. In contrast, CAOS uses a different prediction set construction, together with a monotonicity-based argument, to recover valid finite-sample $1-\alpha$ coverage despite adaptive aggregation and data reuse.
>
> As a sanity check, we also implemented a jackknife+ baseline using the same aggregated nonconformity score as CAOS but with the sensitivity adjusted to also target the same coverage guarantee of $1-\alpha$. Consistent with theory, this approach yields more conservative prediction sets: On RAFT, at $\alpha = 0.1$, averaged over the expanded set of RAFT experiments with 4 LLMs, jackknife+ achieves average coverage of 0.95 vs. 0.91 for CAOS, and produces larger prediction sets (≈7% larger than CAOS on average). This highlights that existing leave-one-out/data-reuse methods at adjusted sensitivity lead to conservative calibration, whereas CAOS leverages a monotonicity-based argument to recover valid marginal coverage at $1-\alpha$. We will clarify this distinction in the revision.

---

> > ### Author Rebuttal · Reviewer_76Fx · 2026-04-02
> >
> > Thank you to the authors for the detailed response. It addresses all of my concerns, and I will maintain my initial positive rating of 5 (Accept).

---

> > > ### Author Response · Authors · 2026-04-08
> > >
> > > Thank you for your thoughtful and constructive feedback. We are glad that our response was able to address your concerns, and we truly appreciate your careful reading and engagement with the methodological and theoretical aspects of our work. Your questions on aggregation baselines and connections to leave-one-out methods were particularly insightful, and we will incorporate these clarifications in the final version.

---

### Official Review · Reviewer_CyE4 · 2026-03-12

**Soundness:** 3
**Presentation:** 2
**Significance:** 3
**Originality:** 3
**Overall Recommendation:** 4
**Confidence:** 3

**Summary:**

This paper studies conformal prediction in the one-shot regime, where each labeled example induces a one-shot predictor, and different predictors may be useful for different test inputs. The paper argues that standard split conformal methods are statistically inefficient in this setting due to data splitting and reliance on a single predictor. To address this, it proposes CAOS, which aggregates one-shot nonconformity scores across labeled examples by summing the smallest k scores, and calibrates the resulting score via a leave-one-out procedure. The main technical contribution is a finite-sample marginal coverage guarantee, proved by reduction to a theoretical full conformal variant. Experiments on one-shot landmark transfer and one-shot text classification suggest improved efficiency over split conformal baselines at comparable coverage.

**Compliance With Llm Reviewing Policy:**

Affirmed.

**Final Justification:**

Some issues were difficult to fully resolve during the response phase; however, the paper's main contributions remain unchanged. I am maintaining my original score.

**Key Questions For Authors:**

1.	The paper fixes k=3 in the main experiments. As the number of reference examples n increases, should the optimal k scale with n, or is the method inherently relying on a small constant number of supportive experts? Have the authors also considered adaptive alternatives, e.g., selecting experts based on score thresholding rather than a fixed k?

2.	Could the authors provide a clearer discussion of calibration and inference complexity as functions of the number of labeled examples and candidate outputs, especially for the LLM-based instantiation?

**Limitations:**

yes

**Strengths And Weaknesses:**

1.	The paper provides a novel monotonicity-based reduction to establish finite-sample marginal coverage for CAOS. By constructing a theoretical full conformal variant as an analytical bridge, the authors transfer coverage guarantees to the leave-one-out procedure.

2.	CAOS demonstrates a favorable efficiency–coverage trade-off, consistently producing smaller prediction sets than SCOS while maintaining empirical coverage near the target level. This makes the method practically meaningful.

Weaknesses:
1.	Although CAOS is much more practical than a theoretical full conformal construction, its computational cost is still nontrivial. Leave-one-out calibration requires aggregated score computation for each labeled example, and test-time inference further scales with both the number of candidate outputs and the number of example-induced predictors. This may limit scalability when the reference set is large.

2.	While summing the smallest k scores is intuitive, the justification for this aggregation rule remains largely empirical. The paper does not explain why this aggregation should be preferred over other strategies, such as min-based pooling, or soft weighting across experts.

3. The experiments in the paper are too simplistic. Perhaps validation should be carried out on larger-scale experiments, such as large-scale image classification or VQA tasks.

4. The idea of leveraging uncertainty for classifier fusion has been explored quite early in the literature, and this paper should include an appropriate discussion and acknowledgment of such prior work — for instance, in the context of Dempster-Shafer evidence theory[1] and uncertainty-aware multimodal fusion [2,3] methods grounded in evidential reasoning.

[1] Upper and lower probabilities induced by a multivalued mapping

[2] Trusted multi-view classification with dynamic evidential fusion

[3] Trustworthy Multimodal Regression with Mixture of Normal-inverse Gamma Distributions

---

> ### Author Rebuttal · Authors · 2026-03-28
>
> We thank the reviewer for the thoughtful and constructive feedback, and for recognizing the novelty of the monotonicity-based approach and the favorable efficiency–coverage trade-off.
>
> ### **Computational cost and scalability**
>
> We agree that a clearer discussion of computational complexity is important and the following discussion will be included in the revision.
>
> Let $N$ denote the number of labeled examples and $L$ the number of candidate outputs. For a single test example, CAOS requires $O(N^2+NL)$ score evaluations, while a single SCOS predictor requires $O(N+L)$. The aggregation step introduces an additional $O(\log k)$ factor, but since $k$ is a small constant (we use $k=3$), this does not affect the overall scaling. For $M$ test examples, this scales to $O(N^2 + MNL)$ for CAOS and $O(N+ML)$ for SCOS. The additional cost of CAOS arises from evaluating all possible reference predictors whereas SCOS considers only a single reference.
>
> In practice, the dominant cost is nonconformity score computation (i.e., LLM forward passes), while conformal post-processing is negligible. On LLaMA-2-7B, the per-fold time was **49.2 s** for CAOS versus **1.0 s** for SCOS. In contrast, in the landmarking setting, embeddings are computed once per image (so all considered methods require the same number of foundation model calls) and subsequent computation reduces to cosine similarity and aggregation, making the practical overhead substantially smaller.
>
> CAOS indeed introduces a statistical–computational tradeoff, exchanging additional computation for improved efficiency via full data reuse. In high-risk settings with scarce labeled data (e.g. clinical applications with expensive annotations), this tradeoff is often favorable.
>
>
> ### **Choice of k and aggregation strategy**
>
> Empirically, CAOS works well with a small constant $k$, consistent with the intuition that only a small subset of reference examples needs to be strongly informative for a given test point. The appendix includes an ablation showing stable performance across a wide range of $k$, with $k=1$ performing slightly best on landmarking. We fixed $k=3$ throughout to avoid introducing task-specific hyperparameter tuning and to keep the method uniform across settings.
>
> These results suggest that CAOS works well with a small constant $k$ in the low-data regime we consider. Motivated by real world challenges, our focus is on the finite-sample, small-$n$ setting; whether scaling $k$ becomes beneficial in larger-data regimes is an interesting open question.
>
> Developing alternative adaptive aggregation schemes such as threshold-based selection are a promising direction for future work. The key challenge is to ensure that they preserve the monotonicity structure required by our analysis.
>
> ### **Relation to prior work on uncertainty-aware fusion**
>
> We thank the reviewer for highlighting connections to uncertainty-aware aggregation and evidential fusion [1] methods. Prior work (e.g., Dempster–Shafer theory–based approaches such as TMC [2] and MoNIG [3]) models predictor uncertainty explicitly (e.g., via Dirichlet or Normal–Inverse-Gamma distributions) and performs adaptive, sample-specific aggregation based on learned uncertainty estimates.
>
> Our work is complementary but focuses on a fundamentally different regime: one-shot prediction with full data reuse, where the same small labeled dataset is used both to construct predictors and to calibrate them, without a clean separation between training and calibration. In this setting, learned uncertainty estimates may be unreliable due to data scarcity and the statistical dependence induced by data reuse.
>
> Our contribution is to show that, even in this setting, it is possible to perform adaptive aggregation with valid finite-sample coverage guarantees, by designing a conformal procedure that operates directly on nonconformity scores and leverages a monotonicity-based argument. There is an interesting opportunity for future work to develop aggregated nonconformity scores that incorporate ideas from evidential learning while still satisfying the monotonicity property required for our guarantees. We will add a discussion of these related approaches in the revision.

---

> > ### Author Rebuttal · Reviewer_CyE4 · 2026-04-04
> >
> > The author addressed my concerns, so I am maintaining my original rating.

---

> > > ### Author Response · Authors · 2026-04-08
> > >
> > > Thank you for your thoughtful and constructive feedback. We are glad that our response was able to address your concerns, and we appreciate your careful reading and engagement with the paper.

---

### Official Review · Reviewer_rg2t · 2026-03-13

**Soundness:** 3
**Presentation:** 4
**Significance:** 3
**Originality:** 3
**Overall Recommendation:** 4
**Confidence:** 3

**Summary:**

This paper introduces Conformal Aggregation of One-Shot Predictors (CAOS), a novel framework designed to provide principled uncertainty quantification for foundation models in data-scarce, one-shot learning scenarios. To overcome the statistical inefficiency of standard split conformal methods that require disjoint data splits, CAOS adaptively aggregates nonconformity scores across multiple one-shot predictors and employs a leave-one-out calibration scheme to maximize the use of available labeled data. Even though this algorithmic approach violates classical exchangeability assumptions at the score level, the authors successfully establish valid finite-sample marginal coverage guarantees through a unique monotonicity-based reduction to full conformal prediction. Ultimately, empirical evaluations on both vision (exemplar-based facial landmarking) and language (RAFT text classification) tasks demonstrate that CAOS maintains reliable target coverage while yielding substantially smaller, more informative prediction sets compared to traditional split conformal baselines.

**Compliance With Llm Reviewing Policy:**

Affirmed.

**Key Questions For Authors:**

1.Could you provide a comparison of the computational cost and inference latency between CAOS and the Split Conformal One-shot Predictor (SCOS) baseline?
2.The facial landmarking task is framed as a patch-based classification task. Would the monotonicity property and CAOS theoretical guarantees (Lemma 5.2 and Theorem 4.1) still hold if the task were natively modeled as a continuous regression problem?
3.CAOS aggregates the k smallest nonconformity scores. How sensitive is this specific summation mechanism to mislabeled or highly noisy reference examples compared to other robust aggregation strategies?

**Limitations:**

See Key Questions

**Strengths And Weaknesses:**

The paper addresses a critical problem of providing principled uncertainty quantification for foundation models in one-shot learning scenarios. This contribution is highly relevant and valuable for high-stakes domains with scarce labeled data, such as the medical facial landmarking application discussed by the authors. The authors propose CAOS, an innovative framework that adaptively aggregates nonconformity scores across multiple one-shot predictors. The paper mentions computational efficiency compared to full conformal prediction, but it lacks a thorough quantitative analysis of the inference latency of CAOS compared to the standard Split Conformal baselines. Aggregating scores across multiple predictors could introduce significant overhead when dealing with large label spaces.
The empirical evaluation on facial landmarking models the task as a 168-class discrete classification problem via patch similarity. It remains unclear how well the CAOS framework would generalize to continuous regression tasks.

---

> ### Author Rebuttal · Authors · 2026-03-28
>
> We thank the reviewer for the thoughtful and constructive feedback, and for recognizing the importance of uncertainty quantification in one-shot settings as well as the novelty and practical relevance of our aggregation framework.
>
> We agree that including a discussion of computational cost, the generalizability of CAOS to the regression setting, as well as the robustness of our aggregation strategy will benefit the paper. The following points will be included in the revision.
>
> ### **Computational cost and scalability**
>
> Let $N$ denote the number of labeled examples and $L$ the number of candidate outputs. For a single test example, CAOS requires $O(N^2+NL)$ score evaluations, while a single SCOS predictor requires $O(N+L)$. The aggregation step introduces an additional $O(\log k)$ factor, but since $k$ is a small constant (we use $k=3$), this does not affect the overall scaling. For $M$ test examples, this scales to $O(N^2 + MNL)$ for CAOS and $O(N+ML)$ for SCOS. The additional cost of CAOS arises from evaluating all possible reference predictors whereas SCOS considers only a single reference.
>
> In practice, the dominant cost is nonconformity score computation (i.e., LLM forward passes), while conformal post-processing is negligible. On LLaMA-2-7B, the per-fold time was **49.2 s** for CAOS versus **1.0 s** for SCOS. In contrast, in the landmarking setting, embeddings are computed once per image (so all considered methods require the same number of foundation model calls) and subsequent computation reduces to cosine similarity and aggregation, making the practical overhead substantially smaller.
>
> CAOS indeed introduces a statistical–computational tradeoff, exchanging additional computation for improved efficiency via full data reuse. In high-risk settings with scarce labeled data (e.g. clinical data with expensive annotations), this tradeoff is often favorable.
>
> ### **Extension to regression settings**
>
> The CAOS framework also applies in the regression setting, when the output space of the individual one-shot predictors is continuous. The key is to still use the same aggregation strategy on the individual regression-style non-conformity scores, such that the CAOS nonconformity score satisfies the monotonicity property in Lemma 5.2.
>
> The main requirements for our theoretical guarantees are symmetry and the monotonicity property, which holds independently of whether the output space is discrete or continuous. We focused our empirical evaluation on classification-style tasks because they represent the most established one-shot prediction settings with current foundation models. Extending these ideas to regression depends on the availability of suitable one-shot regression predictors in a given domain, but the theoretical framework carries over directly.
>
> There is one caveat in regression settings with black-box models: constructing prediction sets typically requires evaluating the nonconformity score over candidate outputs, which in practice is done via discretization or numerical approximation of the label space. This is a general challenge in conformal prediction for regression with complex models and is not specific to CAOS.
>
> ### **Robustness and choice of aggregation (k smallest scores)**
>
> We appreciate the reviewer’s question regarding the aggregation rule. The use of the k-smallest scores is motivated both theoretically and empirically. From a theoretical perspective, this aggregation yields a monotone score function, which is essential for our analysis (Lemma 5.2) and underlies the finite-sample coverage guarantee; this argument applies equally to continuous nonconformity scores.
>
> Empirically, we find that CAOS is not sensitive to the choice of k. The appendix includes an ablation showing stable performance across a wide range (e.g., k=1 to k=10), with k=1 performing slightly best on landmarking. We fixed k=3 throughout the paper to avoid introducing dataset-specific hyperparameter tuning.
>
> We further analyze sensitivity to k by stratifying landmarking tasks by difficulty (measured via SCOS-Oracle performance). CAOS remains stable across k even on more difficult, visually ambiguous landmarks (e.g., regions with many similar patches such as cheeks or forehead), and we observe that the largest gains over SCOS occur precisely in these harder cases. This is consistent with the intuition that aggregating the k smallest scores acts as a form of robust aggregation, focusing on the most informative references while downweighting less relevant ones.
>
> While min pooling is a special case of our aggregation scheme with k=1, we note that alternative aggregation strategies (e.g. median or trimmed means, or soft weighting) are possible, but many do not preserve the monotonicity property required by our analysis and therefore do not directly admit the same finite-sample guarantees. Developing more flexible aggregation schemes that retain this structure is an interesting direction for future work.

---

### Official Review · Reviewer_kUHt · 2026-03-13

**Soundness:** 4
**Presentation:** 3
**Significance:** 3
**Originality:** 3
**Overall Recommendation:** 5
**Confidence:** 3

**Summary:**

This paper proposes conformal aggregation of one-shot predictors (CAOS), a novel conformal prediction method for one-shot prediction.
In CAOS, the score of a given input-output pair on a calibration dataset is defined as the sum of the $k$ smallest nonconformity scores.
Then, the empirical quantile is computed using the leave-one-out method, and the prediction set is output.
Furthermore, while CAOS scores are unexchangeable, the authors provide a coverage guarantee for CAOS by considering the full CAOS score.
The authors verify the effectiveness of the proposed method through experiments using real-world data.

**Compliance With Llm Reviewing Policy:**

Affirmed.

**Final Justification:**

All my concerns were addressed. So I updated my score to 5.

**Key Questions For Authors:**

1. In the RAFT experiment, with a test size of 10 for each task, isn’t the coverage evaluation rather unstable? I believe that the persuasiveness of the results would be enhanced if they were revalidated using a larger evaluation set or a different few-shot classification benchmark.
2. Did you conduct experiments using other models for each task? I thought that if a model performs well enough to consistently assign high values to the nonconformity scores of irrelevant samples, the benefits of the proposed method would be limited.

**Limitations:**

yes

**Strengths And Weaknesses:**

## Strength
- The problem definition and motivation are clear. The authors address an important problem.
- Since one-shot prediction may include many irrelevant samples, the approach of using only $k$ smallest values seems reasonable.
- The paper successfully overcomes the technical challenge that CAOS scores are unexchangeable to derive a coverage guarantee. The approach is valuable and interesting.
- The given dataset is used for both reference and calibration. This is an inherently superior design in settings such as one-shot prediction.

## Weakness
- Even practical CAOS requires $O(nK)$ score evaluation. While this can be a critical issue for LLMs or large label spaces, there is little discussion regarding computational complexity.
- In the RAFT dataset, there are only 10 test data points for each task. This makes the results unreliable.
- The authors conducted their experiments using only a single LLM for each task.

---

> ### Author Rebuttal · Authors · 2026-03-28
>
> We thank the reviewer for the thoughtful and constructive feedback, and for recognizing both the importance of the one-shot setting and the technical contribution of establishing coverage guarantees despite non-exchangeable scores.
>
> ### **Additional experiments with multiple LLMs**
>
> We agree that evaluating across multiple models further strengthens the empirical evidence. In response, we extended the RAFT experiments to additional LLMs (Qwen-32B, Qwen-4B, LLaMA-2-7B, LLaMA-3.2-3B) and repeated all experiments over 5 folds.
>
> Across all models and tasks, we observe consistent trends: CAOS maintains target coverage while yielding smaller prediction sets than SCOS. Averaged across tasks, CAOS reduces set size by 6–12% at $\alpha$ = 0.1 and ~6% at $\alpha$ = 0.2, while maintaining coverage close to nominal (e.g., 0.90–0.92 at $\alpha$ = 0.1, 0.80–0.82 at $\alpha$ = 0.2).
>
> For example, with Qwen-4B at $\alpha$ = 0.1, **CAOS reduces set size by 10.3%** on average while achieving coverage 0.916 vs. 0.902 for SCOS. Similar trends hold across all models tested. We will include these additional results (with confidence intervals over repeated runs) in the revision.
>
> Importantly, these gains persist even when the base model is strong: in tasks where SCOS already produces prediction sets close to size 1, CAOS still achieves consistent additional reductions, indicating that the benefit is not limited to weaker models but arises from more efficient aggregation and data reuse.
>
> ### **Stability of evaluation with small test size**
>
> We emphasize that the RAFT benchmark is intentionally designed for low-data, few-shot evaluation, which aligns directly with the one-shot setting studied in this work. In many real-world text annotation tasks, labeled data is inherently scarce, and RAFT provides a realistic and challenging testbed for this regime.
>
> We agree that evaluating coverage on small test sets (e.g., 10 samples per task) can introduce variance. To address this, we repeated experiments across multiple folds and report mean performance over 5 runs.
>
> The results remain consistent: CAOS maintains valid coverage while achieving improved efficiency over SCOS. We will include these repeated evaluations and variance estimates in the revision to strengthen robustness.
>
> ### **Computational cost and scalability**
>
> We thank the reviewer for the suggestion and will provide a quantitative discussion of computational complexity. CAOS introduces a statistical–computational tradeoff, exchanging additional computation for improved set-size efficiency via full data reuse. This leads to the following computational overhead:
>
> Let $N$ denote the number of labeled examples and $L$ the number of candidate outputs. For a single test example, CAOS requires $O(N^2+NL)$ score evaluations, while a single SCOS predictor requires $O(N+L)$. The aggregation step introduces an additional $O(\log k)$ factor, but since $k$ is a small constant (we use $k=3$), this does not affect the overall scaling. For $M$ test examples, this scales to $O(N^2 + MNL)$ for CAOS and $O(N+ML)$ for SCOS. The additional cost of CAOS arises from evaluating all possible reference predictors whereas SCOS considers only a single reference.
>
> In practice, the dominant cost is nonconformity score computation (i.e., LLM forward passes), while conformal post-processing is negligible. On LLaMA-2-7B, the per-fold time was **49.2 s** for CAOS versus **1.0 s** for SCOS. In contrast, in the landmarking setting, embeddings are computed once per image (so all considered methods require the same number of foundation model calls) and subsequent computation reduces to cosine similarity and aggregation, making the practical overhead substantially smaller.

---

> > ### Author Rebuttal · Reviewer_kUHt · 2026-04-04
> >
> > I thank the authors for their responses. The authors have addressed my concerns. I will update my score.

---

> > > ### Author Response · Authors · 2026-04-08
> > >
> > > Thank you for your thoughtful and constructive feedback. We are glad that our response was able to address your concerns, and we truly appreciate your careful reading and engagement with our work. We will incorporate the runtime analysis and additional experimental results in the final version.

---

### Decision · Program_Chairs · 2026-04-30

**Decision:**

Accept (regular)

**Comment:**

The paper proposes a conformal prediction framework for one-shot settings that aggregates multiple one-shot predictors via a leave-one-out scheme, achieving valid coverage despite breaking exchangeability and improving efficiency over split conformal .

The submission received two accepts and two weak accepts. Overall, reviewers agree the paper is technically sound, and the rebuttal addressed most concrete concerns, including computational cost, additional experiments, and clarification of connections to prior work.

Strengths include a well-motivated problem (uncertainty in one-shot prediction), a novel monotonicity-based argument to recover coverage under non-exchangeable scores, and consistent empirical improvements in prediction set size while maintaining coverage.

The main concerns are limited experimental scope, computational overhead, and some missing comparisons (e.g., aggregation baselines or broader settings), though these were partially addressed in rebuttal.

In summary, given technical contributions and overall positive reviewer sentiment, I recommend acceptance.